# A divergent cyclin/cyclin-dependent kinase complex controls the atypical replication of a malaria parasite during gametogony and transmission

Aurélia C Balestra[1†], Mohammad Zeeshan[2‡], Edward Rea[2‡], Carla Pasquarello[1‡], Lorenzo Brusini[1,2], Tobias Mourier[3], Amit Kumar Subudhi[3], Natacha Klages[1], Patrizia Arboit[1], Rajan Pandey[2], Declan Brady[2], Sue Vaughan[4], Anthony A Holder[5], Arnab Pain[3], David JP Ferguson[4,6], Alexandre Hainard[1], Rita Tewari[2§*], Mathieu Brochet[1§*]

[1]University of Geneva, Faculty of Medicine, Geneva, Switzerland; [2]University of Nottingham, School of Life Sciences, Nottingham, United Kingdom; [3]Biological and Environmental Sciences and Engineering (BESE) Division, King Abdullah University of Science and Technology, Thuwal, Saudi Arabia; [4]Oxford Brookes University, Department of Biological and Medical Sciences, Oxford, United Kingdom; [5]The Francis Crick Institute, Malaria Parasitology Laboratory, London, United Kingdom; [6]University of Oxford, John Radcliffe Hospital, Nuffield Department of Clinical Laboratory Science, Oxford, United Kingdom

**\*For correspondence:**
rita.tewari@nottingham.ac.uk (RT);
Mathieu.Brochet@unige.ch (MB)

[†]These authors contributed equally to this work
[‡]These authors also contributed equally to this work
[§]These authors also contributed equally to this work

**Abstract** Cell cycle transitions are generally triggered by variation in the activity of cyclin-dependent kinases (CDKs) bound to cyclins. Malaria-causing parasites have a life cycle with unique cell-division cycles, and a repertoire of divergent CDKs and cyclins of poorly understood function and interdependency. We show that *Plasmodium berghei* CDK-related kinase 5 (CRK5), is a critical regulator of atypical mitosis in the gametogony and is required for mosquito transmission. It phosphorylates canonical CDK motifs of components in the pre-replicative complex and is essential for DNA replication. During a replicative cycle, CRK5 stably interacts with a single *Plasmodium*-specific cyclin (SOC2), although we obtained no evidence of SOC2 cycling by transcription, translation or degradation. Our results provide evidence that during *Plasmodium* male gametogony, this divergent cyclin/CDK pair fills the functional space of other eukaryotic cell-cycle kinases controlling DNA replication.

## Introduction

Progression through the cell cycle critically relies upon post-translational mechanisms including changes in activity of cell cycle kinases and phosphatases, and ubiquitin-mediated degradation of specific components once their function is complete. Central components of these networks are the cyclin-dependent kinases (CDKs). Many different families of CDK exist (*Lim and Kaldis, 2013*). In most model organisms however, those primarily regulating DNA replication and mitosis belong to a monophyletic family that includes human CDK1-CDK4/6, Cdc2 and Cdc28 in fission and budding yeasts, respectively, and CDKA in plants (*Guo and Stiller, 2004*; *Cao et al., 2014*). In malaria parasites, seven CDKs have been described (*Talevich et al., 2012*; *Tewari et al., 2010*; *Solyakov et al., 2011*). Of these, five are divergent CDK-related kinases (CRK), i.e. they are clearly related to CDKs but have no clear orthologues in the yeast or human kinomes. These include CRK5

(PBANKA_1230200), which forms a distinct branch within the CDK cluster with atypical sequence motifs in the activation loop and an additional 148 amino acid C-terminal extension found only in *Plasmodium* (*Dorin-Semblat et al., 2013*). The gene is required, but not essential, for proliferation of asexual blood stages of the human parasite *P. falciparum* (*Dorin-Semblat et al., 2013*).

The primary regulator of CDK activity is the cyclin subunit. Cyclins were originally named because of their oscillation in level that reach a threshold required to drive cell-cycle transitions (*Morgan, 1995*; *Malumbres and Barbacid, 2009*). Cyclins are now defined as a family of evolutionarily related proteins encoding a cyclin box motif that is required for binding to the CDK catalytic subunit (*Cao et al., 2014*). *Plasmodium* genomes contain no sequence-identifiable $G_1$-, S-, or M-phase cyclins, and only three proteins have sequence homology with cyclin family members in other eukaryotes (*Merckx et al., 2003*; *Roques et al., 2015*). Cyc1 is important for cytokinesis in *P. falciparum* blood stage replication, possibly regulating the CDK7 homolog MRK (*Robbins et al., 2017*). *P. berghei* Cyc3 is dispensable for blood-stage replication but important for oocyst maturation in the mosquito midgut (*Roques et al., 2015*). The paucity of putative cyclins and the diversity of CDKs has led to suggestions that some of these kinases function without a cyclin partner (*White and Suvorova, 2018*).

The malaria parasite has several proliferative phases in its life cycle. Male gametogony or gametogenesis is a proliferative sexual stage in the mosquito vector that is essential for parasite transmission. Circulating mature male gametocytes in the vertebrate host are arrested in a $G_0$-like phase and resume development in the mosquito midgut following a blood meal, activated by the presence of xanthurenic acid (XA) and a drop in temperature (*Billker et al., 1998*). In about ten minutes the haploid male gametocyte completes three rounds of genome replication and closed endomitosis, assembles the component parts of eight axonemes, and following nuclear division, produces eight flagellated motile male gametes in a process called exflagellation (*Billker and Alano, 2005*). The organisation and regulation of the cell cycle during male gametogony is unclear. Current evidence suggests that certain canonical cell-cycle checkpoints are absent (*Alvarez and Suvorova, 2017*). For example, compounds that interfere with mitotic spindle formation do not prevent DNA replication from proceeding (*Billker et al., 2002*; *Zeeshan et al., 2019*), while spindle formation is not affected in a mutant that is unable to replicate DNA (*Zeeshan et al., 2019*; *Fang et al., 2017*). Recently, we observed that proteins involved in DNA replication and cytoskeletal reorganisation are similarly phosphorylated during the first seconds of gametogony (*Invergo et al., 2017*). Interestingly, CRK5 was linked to both groups of proteins, suggesting it has a key regulatory role during male gametogony (*Invergo et al., 2017*). Here, we provide evidence suggesting CRK5 is part of a unique and divergent CDK/cyclin complex required for progression through male gametogony and essential for parasite transmission.

## Results

### CRK5 is a key regulator of gametogony and sporogony in the mosquito

Previous attempts to disrupt *P. berghei crk5* had suggested the gene is essential for asexual blood-stage proliferation (*Tewari et al., 2010*). However, using long sequence homology regions to replace *crk5* with a *T. gondii* DHFR/TS resistance marker (*Figure 1A* and *Figure 1—figure supplement 1A*), we obtained resistant parasites and cloned them following a two-step enrichment. The gene deletion in the resulting CRK5-knockout (KO) clone was confirmed by PCR (*Figure 1—figure supplement 1A*) and RNAseq analysis (*Figure 1A* and *Supplementary file 1*). There was no significant growth defect during erythrocytic asexual multiplication (*Figure 1B*), nor an inability to produce morphologically normal gametocytes (*Figure 1C*). However, upon XA activation only a few microgametocytes formed active exflagellation centres (*Figure 1D*). While no major transcriptional changes were detected (*Figure 1E*), a significant (p-value<$10^{-2}$) but limited increase in expression (0 < log$_2$[-Fold Change]<1) of multiple regulators of gametogony (including *cdpk4*, *map2k*, *nek1* and *ark2*) was observed in CRK5-KO gametocytes (*Supplementary file 1*). This could indicate that CRK5 plays a role during gametocytogenesis prior to activation of gametocytes or that deletion of its coding sequence may be compensated by upregulation of other kinases. Quantitative PCR analysis confirmed these observations but in most cases, changes were not statistically significant (*Figure 1F*). To test the possibility that the exflagellation defect was due to multiple subtle transcriptional

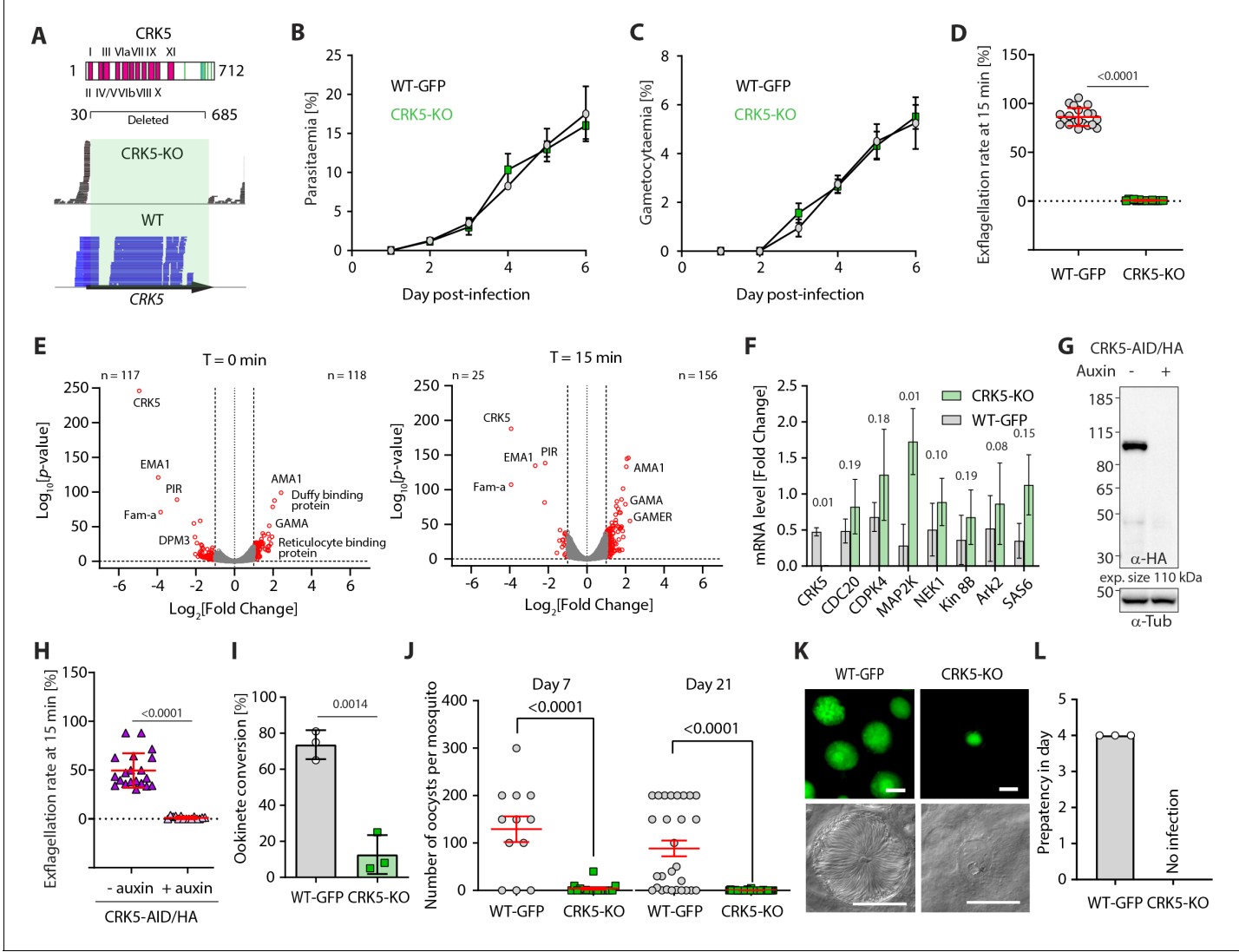

**Figure 1.** CRK5 is a key regulator of gametogony and sporogony during mosquito transmission. (A) Schematic representation of CRK5 showing the eleven regions that constitute the catalytic domain in pink, and the *Plasmodium*-specific C-terminal extension that is phosphorylated at eight sites (green bars). Amino acids 31 to 685 were replaced by the selection marker in the CRK5-KO line. The lower panel shows the sequence reads mapping on *crk5* in the KO and WT lines, confirming deletion of the gene in the KO. (B) Deletion of *crk5* has no effect on the growth of asexual blood stage parasites (error bars show standard deviation from the mean; 3 and 2 independent infections for the CRK5-KO and WT lines, respectively). (C) Deletion of *crk5* has no effect on gametocytaemia (error bars show standard deviation from the mean; 3 and 2 independent infections for the CRK5-KO and WT lines, respectively). (D) Deletion of *crk5* leads to a strong defect in exflagellation as measured by the percentage of active exflagellation centres per microgametocyte (error bars show standard deviation from the mean; technical replicates from three independent infections; two-way ANOVA). (E) Volcano plots show limited transcriptional changes between the CRK5-KO and wild type (WT) parasite lines, comparing non-activated gametocytes (T = 0 min) with gametocytes 15 min after activation (T = 15 min). (F) The levels of mRNA for multiple critical regulators of gametogony do not change significantly in CRK5-KO compared to WT parasites, despite a trend for up-regulation in the mutant (error bars show standard deviation from the mean; two independent biological replicates, two-tailed unpaired t-test). (G) Depletion of CRK5-AID/HA protein upon addition of auxin to mature purified gametocytes, as measured by western blotting; α-tubulin serves as a loading control. (H) Depletion of CRK5-AID/HA upon auxin treatment leads to a profound defect in exflagellation (error bars show standard deviation from the mean; technical replicates from three independent infections; two-way ANOVA). (I) Effect of *crk5* deletion on ookinete formation as a measure of fertile male gamete formation. Conversion is reported as the percentage of female gametes forming ookinetes (error bars show standard deviation from the mean; three independent infections, two-tailed unpaired t-test). (J) *crk5* deletion leads to a dramatic reduction in oocyst formation when mature gametocytes are fed to mosquitoes. Midguts were dissected and GFP-positive oocysts were counted at 7 and 21 days post-infection (error bars show standard deviation from the mean; three independent infections; two tailed unpaired t-test). (K) The residual CRK5-KO oocysts show abnormal development; scale bar = 20 μm. (L) No viable CRK5-KO sporozoite develop. Mosquitoes were infected and 21 days later allowed to feed on naive mice. For the WT line, blood stage parasites were readily observed after a 4 day

*Figure 1 continued on next page*

*Figure 1 continued*

prepatent period, while no mice became infected with CRK5-KO parasites (error bars show standard deviation from the mean; three independent infections).

The online version of this article includes the following source data and figure supplement(s) for figure 1:

**Source data 1.** CRK5 is a key regulator of gametogony and sporogony during mosquito transmission.
**Source data 2.** Panel G, anti-HA western blot.
**Source data 3.** Panel G, anti-αtubulin western blot.
**Figure supplement 1.** Generation of CRK5-KO and CRK5-AID/HA transgenic lines and CRK5-GFP expression in oocysts.

changes prior to activation, we used an additional approach to study the effect of rapid CRK5 degradation in mature gametocytes. We tagged the endogenous *crk5* gene with an AID/HA epitope tag (*Figure 1—figure supplement 1B*) to degrade the fusion protein in presence of auxin in a strain expressing the Tir1 protein (*Philip and Waters, 2015*; *Figure 1G*). Addition of the AID/HA tag to the CRK5 C-terminus imposed a significant fitness cost, with a 2-fold decrease in exflagellation in the absence of auxin, but importantly, depletion of CRK5-AID/HA by one hour of auxin treatment of mature gametocytes prior to activation resulted in a dramatic reduction in exflagellation (*Figure 1H*).

Since a residual number of CRK5-KO male gametocytes produced active exflagellation centres, we asked whether the resulting microgametes were fertile by measuring the number of ookinetes per activated female gametocyte. Consistent with the dramatic reduction in male gametogony, CRK5-KO parasites showed a significant reduction in ookinete conversion compared to the parental control line that expresses GFP under the strong *eef1α* promoter (*Janse et al., 2006a*; *Figure 1I*). This reduction led to a > 30 fold decrease in the number of CRK5-KO oocysts present on the midgut epithelium of *Anopheles stephensi* mosquitoes allowed to feed on infected mice (*Figure 1J*). To further test for a possible role of CRK5 during sporogony, we endogenously tagged CRK5 with GFP at its C-terminus (*Figure 1—figure supplement 1C*). Consistently, CRK5-GFP is expressed in early oocysts (*Figure 1—figure supplement 1D*), which corroborates that residual CRK5-KO ookinetes produced non-viable oocysts (*Figure 1K*), suggesting an additional role during sporogony (*Figure 1L*). Altogether, these results demonstrate a critical requirement for CRK5 during male gametogony and an essential function for CRK5 in parasite development and transmission.

## Phosphoproteome kinetics point to direct phosphorylation of the pre-replicative complex by CRK5

To elucidate the processes regulated by CRK5 during gametogony, we compared the proteomes and phosphoproteomes of WT and CRK5-KO gametocytes using TMT labelling. We focus our analysis on the first minute of gametogony that covers the first S-phase and metaphase of mitosis I (*Figure 2A*). Apart from a significant CRK5 reduction in the CRK5-KO line, no significant differences among the 2598 quantified proteins were observed between mutant and parental lines (*Supplementary file 2*). Similarly, non-activated CRK5-KO gametocytes showed limited differences in phosphorylation compared to the parental counterpart. However, out of the 5901 quantified peptides, 20 phosphopeptides showed significant down-regulation upon activation (Benjamini-Hochberg adjusted *p*-values <0.05 - *Figure 2B*) with most enriched GO terms corresponding to proteins important in DNA replication including the origin recognition complex (ORC - *Figure 2C* and *Supplementary file 3*). This group includes a protein with similarities to the licensing factor CDT1 (chromatin licensing and DNA replication factor 1, PBANKA_1356300), a possible orthologue of the DNA replication factor CDC6 (PBANKA_1102900), and two ORC components (ORC2 - PBANKA_0803000 - and ORC4 – PBANKA_1348800). Interestingly, eight down-regulated phosphorylation sites map to the S/T*PxK CDK consensus motif (*Figure 2D*). This motif shows a significant >600 fold enrichment compared to all other phosphorylated sequences identified in this study (p-value<$1.10^{-14}$).

Multiple phosphopeptides were more abundant following activation (*Figure 2B*) suggesting rapid secondary effects on CRK5 phospho-dependent pathways, probably via the regulation of protein phosphatases, as well as the inactivation or non-activation of other kinases. For example, the phosphatase PPM11 in the mutant line showed a significantly different phosphorylation profile before

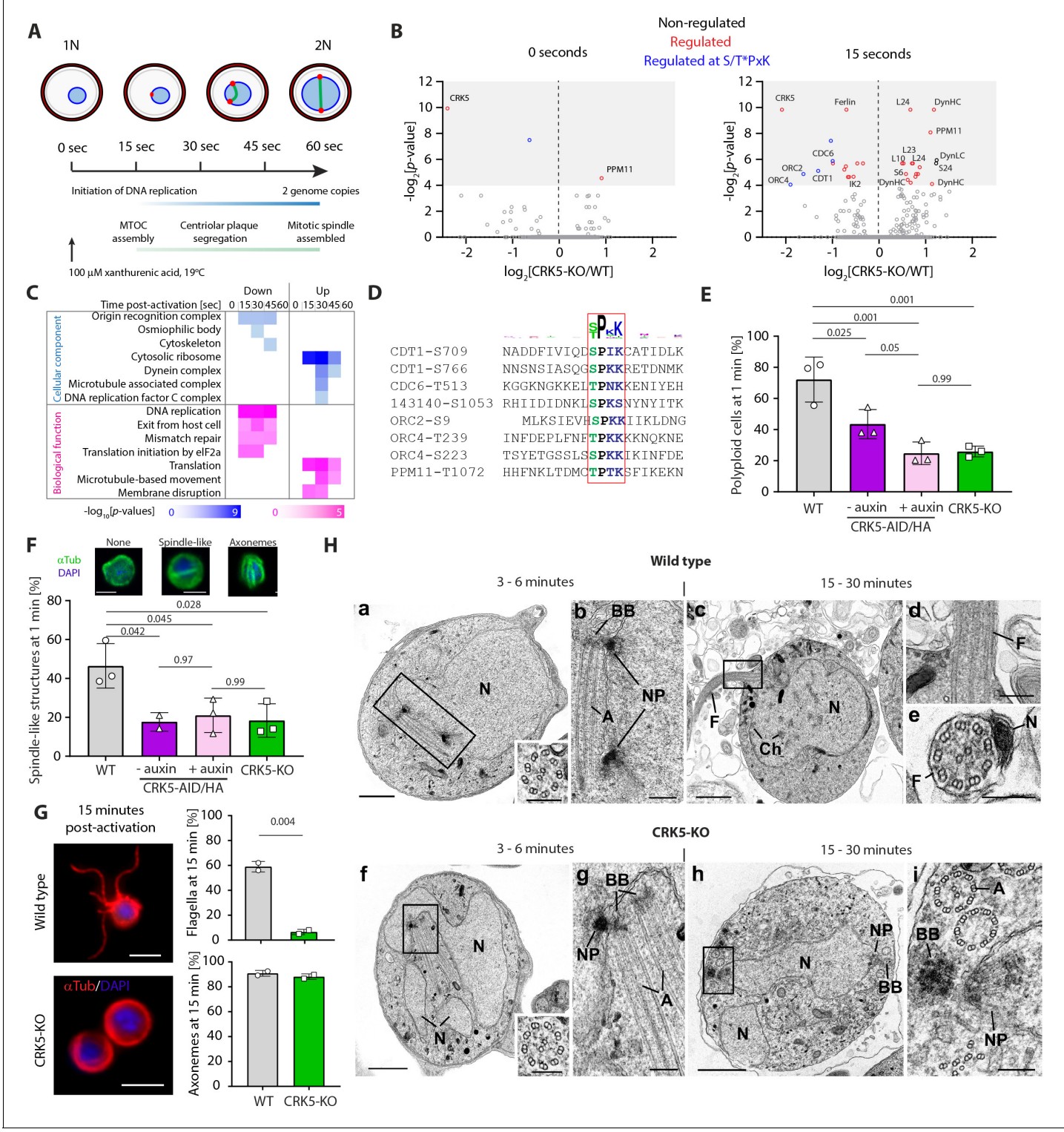

**Figure 2.** CRK5 shows functional similarities to canonical CDKs and directly regulates DNA replication during gametogony. (**A**) Schematic of the first mitosis following activation of male gametogony. Red dots represent the nuclear poles and the green line the mitotic-spindle. Time points selected for a high-time resolution phosphoproteome survey of CRK5-KO parasites are indicated. B. Volcano plots showing the extent of differentially phosphorylated peptides in non-activated and activated gametocytes after 15 s, comparing CRK5-KO and WT parasite lines. Significantly regulated sites (p-value<0.05) with a fold change >2 are highlighted in red, and in blue when corresponding to the CDK S/T*PxK motif. C. GO term enrichment analysis for down- and up-regulated phosphopeptides in CRK5-KO gametocytes during the first minute of gametogony. D. Alignment of eight down-regulated phosphopeptides showing the classical S/T*PxK CDK motif. E. Reduction in the number of male gametocytes replicating their DNA upon

*Figure 2 continued on next page*

*Figure 2 continued*

*crk5* deletion or CRK5-AID/HA depletion. Proportion of male gametocytes undergoing DNA replication was determined at 1 min pa (post-activation) and is expressed as a percentage of polyploid (>1N) cells (error bars show standard deviation from the mean; three independent infections; two-way ANOVA). F. Reduction in the number of male gametocytes with spindle-like structures upon *crk5* deletion or CRK5-AID/HA depletion, as assessed by α-tubulin staining 1 min pa (error bars show standard deviation from the mean; three independent infections; two-way ANOVA). Inset pictures show representative typical tubulin distribution patterns (green), observed in WT microgametocytes counter stained with DAPI (blue); scale bar = 2 μm. G. CRK5-KO gametocytes, fifteen min after activation, have fully assembled axonemes that do not initiate flagellar motility, in contrast to their WT counterparts (error bars show standard deviation from the mean; two independent infections; two tailed unpaired t-test). Scale bar = 5 μm. H. Electron micrographs of male gametogony in wild type parasites at 3–6 min (a, b) and 15–30 min (c, d, e) and CRK5-KO mutant parasites at 3–6 min (f, g) and 15–30 min (h, i). Bars represent 1 μm (a, c, f, h) and 100 nm in other images. (a) Low power micrograph an early gametocyte showing the large nucleus (N) and two nuclear poles. Insert: detail of a cross-sectioned axoneme. (b) Detail of the enclosed area showing the nuclear poles (NP) with an adjacent basal body (BB) and elongated axoneme (A). (c) Late microgametocyte undergoing exflagellation showing the nucleus (N) with some clumped heterochromatin (Ch) and a flagellum (F) protruding from the surface. (d) Detail of the enclosed area showing the point of exit for the microgamete flagellum (F). (e) Cross-section through a free microgamete showing the 9+two flagellum (F) and electron dense nucleus (N). (f) Low power of an early microgametocyte showing an irregular lobated nucleus (N) and the presence of a nuclear pole and axonemes unusually centrally located in the cytoplasm. Insert: detail of a cross-sectioned axoneme. (g) Detail of the enclose area showing the nuclear pole (NP) closely associated with a basal body (BB) and longitudinally running axonemes. (h) At a late time point, the parasite still resembles an early stage with lobated nucleus (N) which displays nuclear poles (NP) and basal bodies (BB). (i) Detail of the enclosed area showing the nuclear pole (NP) and associated cross section basal body (BB). Note the normal and abnormal axonemes (A).

The online version of this article includes the following source data and figure supplement(s) for figure 2:

**Source data 1.** CRK5 shows functional similarities to canonical CDKs and directly regulates DNA replication during gametogony.

**Figure supplement 1.** Percentage of GO terms shared between CRK5 and a set of *Plasmodium* and human kinases.

**Figure supplement 1—source data 1.** Percentage of GO terms shared between CRK5 and a set of *Plasmodium* and human kinases.

and after activation, although the functional relevance of PPM11 phosphorylation remains to be determined. Components of the ribosome were the most significantly enriched (*Figure 2C*), suggesting CRK5 can contribute indirectly to the activation of translation that is de-repressed upon gametocyte activation (*Mair et al., 2006*). Interestingly, four phosphopeptides from dynein subunits were also more abundant, suggesting that the microtubule-based movement required during gametogony may also be dependent on CRK5.

The set of enriched GO terms associated with CRK5-regulation is similar to those associated with calcium dependent protein kinase (CDPK) four and CRK4 (*Figure 2—figure supplement 1*), two kinases previously shown to be crucial for DNA replication and its initiation during gametogony (*Fang et al., 2017*) and schizogony (*Ganter et al., 2017*), respectively. When compared with a set of human kinase-associated GO terms as previously described (*Ganter et al., 2017*), the CRK5 set shows similarities to those of CDK1 and CDK2 as well as other cell-cycle kinases including Chk1, ATM and Aurora B. Altogether this phosphoproteomic analysis shows that components of the pre-replicative complex are phosphorylated on a canonical CDK motif in a CRK5-dependent manner suggesting that CRK5 functions as a CDK. CRK5 may also indirectly regulate phosphorylation of proteins involved in translation and microtubule-based movement.

## CRK5 is required for both S- and M- phases during *P. berghei* gametogony

Given CRK5 similarity to canonical CDKs and its possible role in initiating DNA replication, we quantified the DNA content by flow cytometry of microgametocytes lacking *crk5* or upon CRK5-AID/HA degradation mediated by auxin. In both cases, we observed an average 2-fold fewer polyploid (≥2N) gametocytes 1 min post-activation (pa) with XA (*Figure 2E*), demonstrating a dependency on CRK5 for DNA replication during male gametogony.

It has been suggested that M-phase is initiated independently of S-phase completion during *Plasmodium* gametogony (*Fang et al., 2017*; *Invergo et al., 2017*). As multiple dynein-related proteins are phosphorylated in a CRK5-dependent manner, we quantified the formation of spindle-like structures and axonemes in CRK5-KO parasites. Cells depleted of CRK5 are unable to correctly assemble or maintain the spindle, with a greater than two-fold decrease of visible spindles at 1 min pa (*Figure 2F*) in absence of CRK5. It is important to note that this decrease is also observed in absence of Auxin in the CRK5-AID/HA line suggesting that the defect in exflagellation observed for this line is also associated with a slower gametogenesis. Tubulin staining at 15 min pa showed that axoneme

formation is not affected but axonemal beating is not initiated (*Figure 2G*). Ultrastructural analysis by electron microscopy confirms the typical (9+2) axoneme structure; however at 6 min pa there appears to be more nuclear poles present in WT compared to mutant parasites (0.56/cell compared to 0.32/cell; random sample of individual sections from 50 cells of each population) consistent with reduced or slower mitotic progression in the mutant. The cytoplasm contained electron dense basal bodies, which were often, but not always, associated with nuclear poles (*Figure 2Hb* and g). Deletion of *crk5* appeared to prevent activated microgametocytes developing to form multiple nuclear poles and they showed no chromatin condensation, cytokinesis or flagellum formation (*Figure 2H*).

These results indicate that CRK5 is a critical primary regulator of DNA replication initiation and progression and, possibly, an indirect regulator of microtubule-based processes required for spindle and nuclear pole formation in the nucleus, but not for axoneme assembly in the cytosol.

## CRK5 is part of an atypical nuclear cyclin/CDK complex

The binding of cyclins to CDKs is required for progression through cell-cycle phases in model organisms. In the absence of clear S- and M- phase cyclin homologues, we searched for possible CRK5 regulatory subunits. To identify any proteins bound to CRK5, we used the CRK5-GFP line and additionally tagged the protein with 3xHA at its C-terminus by modification of the endogenous gene (*Figure 3—figure supplement 1A*). The tagged protein was localised to the nucleus in non-activated gametocytes (*Figure 3A*) and at 1 min pa showed an additional location at the mitotic spindle as suggested by co-localisation with tubulin (*Figure 3—figure supplement 1D*). Mass-spectrometry analyses of affinity-purified tagged CRK5-3xHA (0, 15, 30 and 60 sec pa) and CRK5-GFP (at 6 and 15 min pa) identified 234 proteins (*Supplementary file 4*). Consistent with a role for CRK5 in the initiation of DNA replication, GO term enrichment analyses of proteins co-purifying with CRK5 highlighted ORC, minichromosome maintenance (MCM), RepC and alpha DNA polymerase primase complexes (*Figure 3B and C*).

One of the most abundant proteins co-purified with CRK5 was SOC2 (PBANKA_1442200), a protein we had previously found to be important for cell-cycle progression during gametogony (*Fang et al., 2017*). SOC2 has very low homology (<13%) with a cyclin domain in a protein of the starfish *Asterina pectinifera* (*Merckx et al., 2003*), but the protein lacks key residues across most of the cyclin box (*Roques et al., 2015*) and has no detectable cyclin-like function in in vitro biochemical assays (*Merckx et al., 2003*). Analysis of CRK5 immunoprecipitates also identified multiple peptides from PBANKA_0824400 (CDK regulatory subunit, CDKrs), a protein related to Cks1 and CksHs2, two CDK-associated proteins in *Saccharomyces cerevisiae* (*Bourne et al., 2000*) and human (*Parge et al., 1993*), respectively. To confirm the putative interaction between CRK5, SOC2 and CDKrs, we tagged both SOC2 and CDKrs at the C-terminus by integration of 3xHA (*Fang et al., 2017*) or GFP coding sequence into the endogenous gene (*Figure 3—figure supplement 1B*) and affinity purified SOC2-3xHA (0, 15, 30 and 60 sec pa) or CDKrs-GFP (at 60 s, 6 min and 30 min pa). Reciprocal detection of CRK5, SOC2 and CDKrs components was observed (*Supplementary file 4*). In addition, each protein also co-purified the ORC, MCM, RepC and alpha DNA polymerase primase complexes (*Supplementary file 4*).

To better characterise the specificity of CRK5, SOC2 and CDKrs co-immunoprecipitations, we compared the relative abundance of proteins recovered in CRK5, SOC2 and CDKrs immunoprecipitates at 1 min post-activation with those of four proteins involved in DNA replication or gametogenesis. These included CDPK4 that is critical to initiate DNA replication (*Fang et al., 2017*; *Billker et al., 2004*) as well as MCM5. We also used two substrates of CDPK4, SOC1 that is important for DNA replication and SOC3 that is required later in gametogenesis prior to axoneme activation (*Fang et al., 2017*). Seven-dimension principal-component analysis (PCA) of all detected proteins confirmed the clustering of CRK5 with SOC2 and CDKrs consistent with the formation of a complex (*Figure 3C*). As previously proposed, CDPK4 was associated with MCM proteins while ORC and RepC proteins clustered in between the two groups.

Previous bioinformatics analyses had identified only three *bona fide Plasmodium* cyclins (*Roques et al., 2015*). Here, the identification of SOC2 as a putative CRK5 binding partner suggests it is also a cyclin. We revisited the phylogenetic analyses of apicomplexan cyclins and CDKs, and found that CRK5 and CDKrs are conserved in *Haemosporidia*, *Coccidia* and *Piroplasma*, whilst SOC2 is found only in *Haemosporidia* (*Figure 3D* and *Figure 3—figure supplement 1C*).

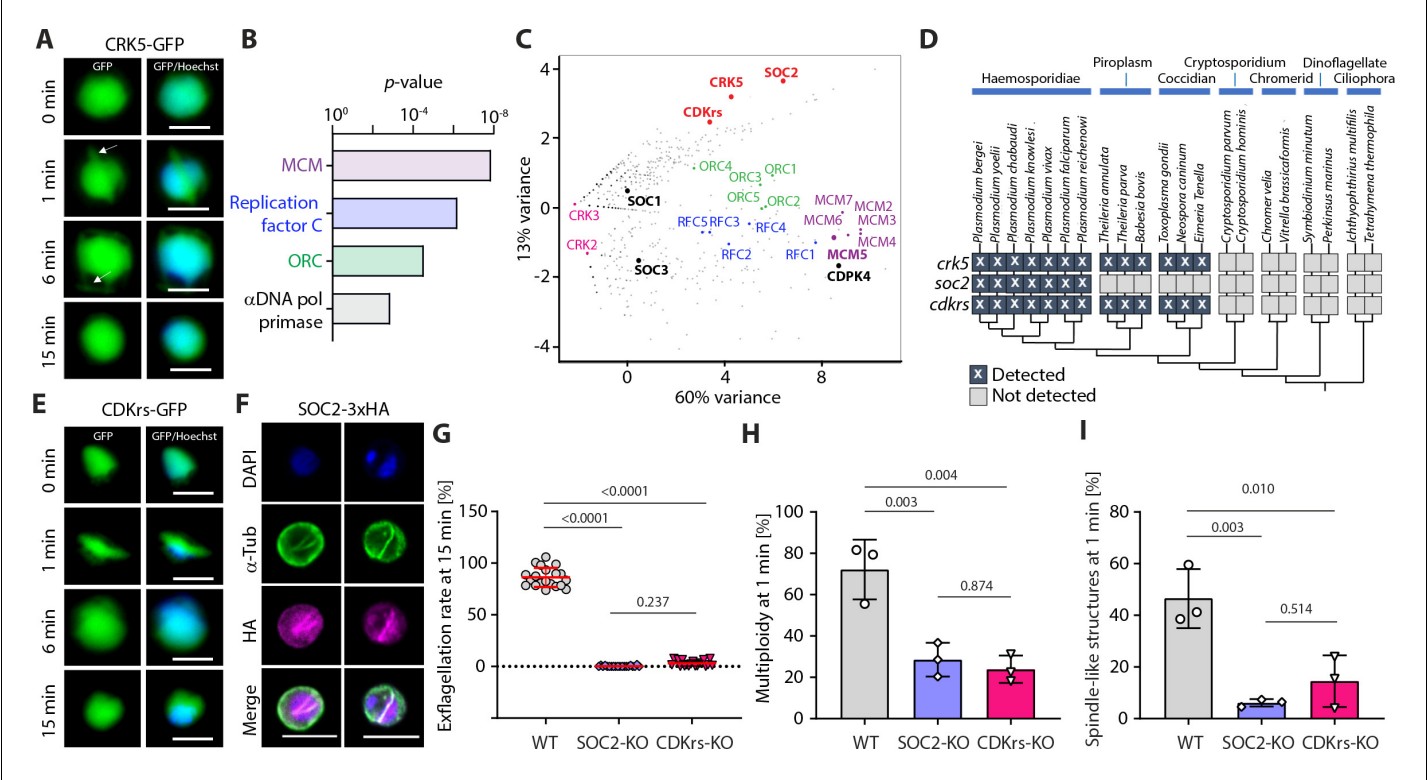

**Figure 3.** CRK5 interacts with a divergent cyclin and a conserved CDK regulatory subunit. (**A**) Localisation of CRK5-GFP during male gametogony. The protein colocalises with Hoechst DNA-binding dye and shows a mitotic spindle distribution during M-phases. Scale bar = 5 μm. (**B**) GO term enrichment analysis of proteins co-immunoprecipitated with CRK5 from gametocytes. Bonferroni corrected *p*-values are indicated. (**C**) Spectral count values as identified by mass spectrometry for proteins co-purifying with CRK5, CDKrs, SOC2, CDPK4, SOC1, SOC3 and MCM5 (highlighted in bold) following immunoprecipitation, and displayed in first and second principal components. Proteomic sets (including CRKs, cyclins, MCMs, ORCs and RepC complex proteins) are additionally highlighted. (**D**) Phylogenetic distribution of the CSC complex components detected in alveolate genomes. Dark boxes with X denote the presence of genes. (**E**) Localisation of CDKrs-GFP during male gametogony. The protein colocalises with Hoechst dye and shows a mitotic spindle distribution during M-phases. Scale bar = 5 μm. (**F**) Localisation of SOC2-3xHA by immunofluorescence in activated male gametocytes. The protein colocalises with DAPI and shows overlap with mitotic spindles stained by α-tubulin antibodies during M-phase. Scale bar = 5 μm. (**G**) Deletions of *soc2* or *cdkrs* lead to a profound defect in exflagellation (error bars show standard deviation from the mean; three independent infections; two-way ANOVA). (**H**) Reduction in the number of male gametocytes replicating their DNA in parasites with *soc2* or *cdkrs* deletions. The proportion of male gametocytes undergoing DNA replication was determined at 1 min pa and is expressed as the percentage of cells that are polyploid (>1N) (error bars show standard deviation from the mean; three independent infections; two-way ANOVA). (**I**) Reduction in the number of male gametocytes with spindle-like structures following *soc2* or *cdkrs* deletion, as assessed by α-tubulin staining 1 min pa (error bars show standard deviation from the mean; three independent infections; two-way ANOVA).

The online version of this article includes the following source data and figure supplement(s) for figure 3:

**Source data 1.** CRK5 interacts with a divergent cyclin and a conserved CDK regulatory subunit.

**Figure supplement 1.** Identification and characterisation of the CSC complex components.

**Figure supplement 1—source data 1.** DNA sequences used in panel D.

**Figure supplement 1—source data 2.** Accession numbers of genes analysed in panel D.

## CRK5, SOC2 and CDKrs have a similar location and complementary functions during gametogony

Given the biochemical evidence that CRK5, SOC2 and CDKrs form a complex, we wanted to compare their subcellular location. Live fluorescence microscopy identified CDKrs-GFP in the nucleus of non-activated microgametocytes and throughout male gametogony, with a spindle-like localisation during mitoses, as found for CRK5 (*Figure 3E*). By indirect immunofluorescence of fixed cells, we confirmed the pattern of SOC2-3xHA distribution (*Figure 3F*), as previously described (*Fang et al., 2017*).

SOC2-KO microgametocytes showed defects in the ploidy transitions during gametogony (*Fang et al., 2017*). In agreement with the notion of complementary roles for CRK5, SOC2 and CDKrs, deletion of either SOC2 and CDKrs encoding genes (*Figure 3—figure supplement 1E*) also resulted in a severe reduction in exflagellation (*Figure 3G*). The defect was already apparent at 1 min pa, with a > 3 fold reduction of cells with ≥2N DNA content (*Figure 3H*) and a 3- fold reduction in mitotic spindles (*Figure 3I*). Ultrastructural analysis by electron microscopy of SOC2-KO and CDKrs-KO activated gametocytes confirms an early arrest during male gametogony as observed in CRK5-KO parasites (*Figure 3—figure supplement 1F*). Altogether, these data strongly suggest that the CRK5/SOC2/CDKrs complex is a functional CDK/cyclin complex that controls early stages of S-phase and possibly controls progression through the following or concurrent M-phase during gametogony. We propose the name 'CSC complex' for this CRK5/SOC2/CDKrs complex.

## SOC2 expression does not follow a temporal cyclin pattern during gametogony and the CSC complex is stable during the first round of mitosis

Cyclin turnover mediated by the ubiquitin-dependent proteasome pathway causes oscillations in CDK activity and therefore we wished to examine the stability of the CSC complex. Western blot and fluorescent microscopy analysis revealed that CRK5, SOC2 and CDKrs are expressed in non-activated gametocytes (*Figures 3A, E* and *4A*) and therefore we examined whether the CSC complex is degraded during the course of gametogony. Western blot analysis of CRK5-3xHA and SOC2-3xHA over the 10 min of gametogony revealed no evidence for degradation of either protein (*Figure 4A*). To confirm that levels of CRK5 and CDKrs are maintained during gametogony in single cells, we monitored by flow cytometry the fluorescence of gametocytes expressing CRK5-GFP or CDKrs-GFP during gametogony. This analysis confirmed that there was no change in the level of CRK5-GFP or CDKrs-GFP fluorescence following male gametocyte activation and gametogony (*Figure 4—figure supplement 1A*).

To examine whether the apparent stable levels of SOC2-3xHA and CRK5-3xHA result from a balance of de novo translation and protein degradation, we pre-treated non-activated gametocytes with the proteasome inhibitor MG132 for one hour. Although MG132 significantly reduced exflagellation (*Figure 4—figure supplement 1B*), it did not affect DNA replication (*Figure 4—figure supplement 1C*) nor lead to a significant accumulation of SOC2, CRK5, or CDKrs as assessed by western blot (*Figure 4A*) and flow cytometry (*Figure 4—figure supplement 1A*). Together, these results suggest no oscillation in levels of the CSC complex resulting from differential temporal translation and degradation during gametogony.

CDKs are usually characterised by low activity in the absence of a bound cyclin, therefore we reasoned that dynamic assembly of the CSC complex might underlie CRK5 regulation. Development of activated gametocytes is highly synchronous and we used this property to investigate the interaction between CRK5 and SOC2 in the first minute after activation. We immunoprecipitated SOC2-3xHA and CRK5-3xHA at 0, 15, 30 and 60 seconds pa and measured the relative abundance of immunoprecipitated proteins in the CSC complex by label-free mass-spectrometry (*Supplementary file 4*). We detected no difference in the levels of SOC2 or CRK5 protein/protein interactions over this time course (*Figure 4B*). However, we noticed that a CDT1-like protein became less abundant in CRK5 and SOC2 immunoprecipitates following gametocyte activation (*Figure 4C*). Pre-treatment of gametocytes with BKI-294, a CDPK4 inhibitor, prevented this reduction. This suggests that the protein is degraded or less accessible to the complex following CDPK4 activation. These results suggest that SOC2-CRK5 binding is not dynamic during the first round of DNA replication.

## CSC is dynamically phosphorylated during the first round of replication

Reversible phosphorylation of cyclin-CDK complexes regulates kinase activity and therefore we examined whether this mechanism may regulate the activity of the CSC complex. We had previously identified 13 phosphorylated SOC2 residues (*Invergo et al., 2017*), with S5088/5089 phosphorylated in a CDPK4-dependent manner during early gametogony (*Fang et al., 2017*). Here, S5088 was also phosphorylated in a CDPK4-dependent manner in the immunoprecipitated SOC2 (*Figure 4—figure supplement 1D*), but S5088/5089 substitution with alanine (*Figure 4—figure supplement 1E*) did not cause a significant defect in DNA replication or exflagellation (*Figure 4—figure*

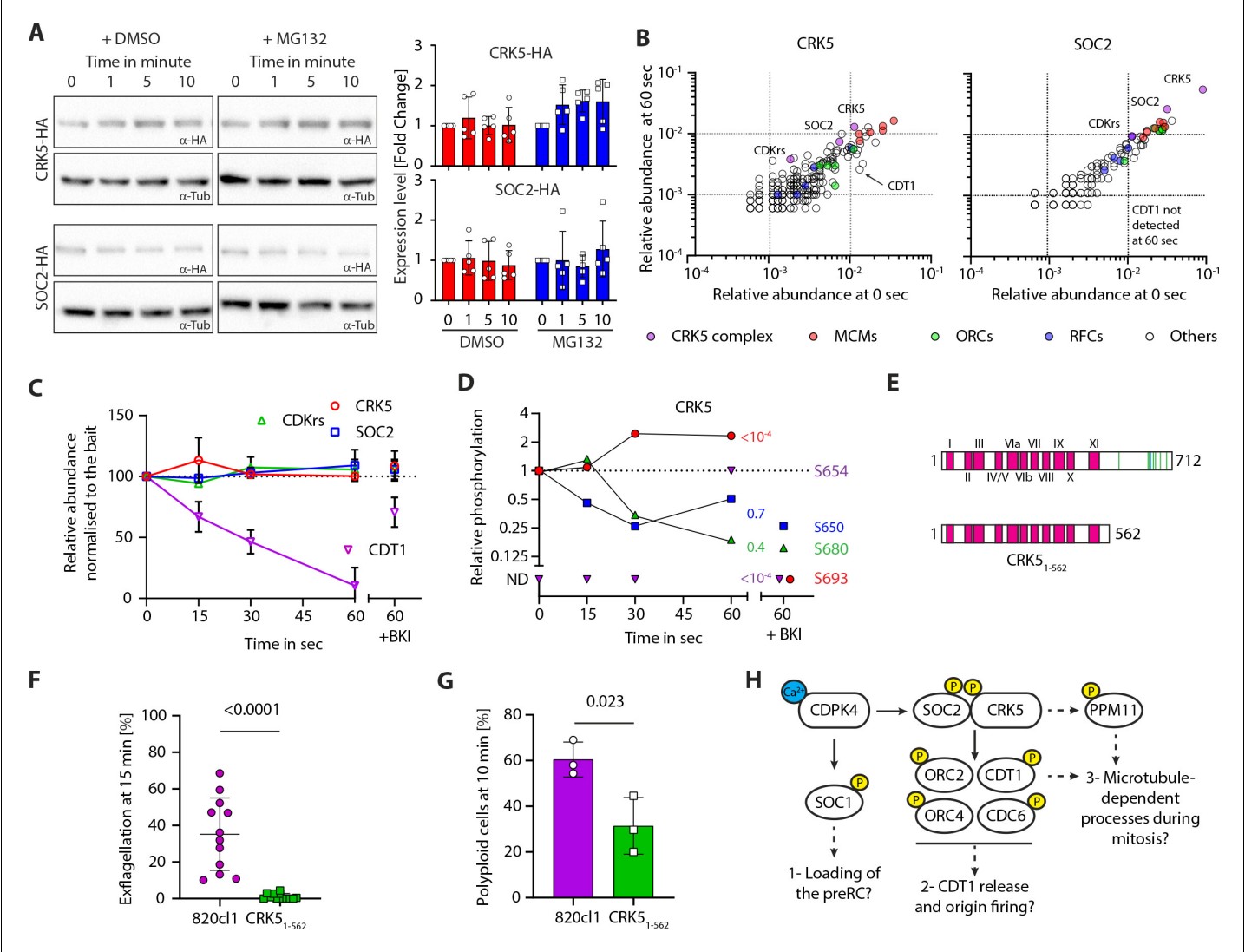

**Figure 4.** The CSC complex is stable but dynamically phosphorylated during gametogony. (A) Western blot analysis of CRK5-3xHA and SOC2-3xHA over the course of gametogony shows that the level of each protein is stable in absence or presence of the proteasome inhibitor MG132; α-tubulin serves as a loading control. Quantification from five independent biological samples is shown to the right (error bars show standard deviation from the mean). (B) Dot blots showing the relative abundance of proteins immunoprecipitated (% of all spectra) from gametocytes at 0 or 60 s activation, with CRK5-3xHA (left panel) and SOC2-3xHA (right panel). No major changes were observed between the two conditions; CDT1 is however recovered less in both IPs at 60 s. (C) Relative abundance of CRK5, SOC2, CDKrs and CDT1 in SOC2-3xHA and CRK5-3xHA IPs over the first min of gametogony and in gametocytes pre-treated with the CDPK4 inhibitor BKI-1294 (*Ojo et al., 2014*) (1 µM). Abundance are relative to the bait and normalised to the 0 s time point (error bars show standard deviations from the mean; data are extracted from single SOC2-3xHA and CRK5-3xHA immunoprecipitations). (D) Phosphorylation profile of immunoprecipitated CRK5 during the first minute of gametogony in presence or absence of the CDPK4 inhibitor BKI-1294 (1 µM). ND = Not detected. (E) Schematic representation of the CRK5$_{1-562}$ mutant generated in this study. (F) CRK5$_{1-562}$ gametocytes show a strong defect in exflagellation (error bars show standard deviation from the mean; three independent infections; two-tailed t-test). (G) Reduction in the number of male gametocytes with a *crk5* 3′ deletion replicating their DNA. The proportion of male gametocytes undergoing DNA replication was determined at 10 min pa and is expressed as a percentage of cells that are polyploid (>1N) (error bars show standard deviation from the mean; three independent infections; two-tailed t-test). (H) Possible signalling cascades regulating entry and progression through the first replicative cycle during *P. berghei* gametogony. Early calcium mobilisation upon gametocyte activation leads to CDPK4-dependent phosphorylation of SOC1 and the CSC complex. SOC1 may be involved in the loading of the pre-replicative complex (preRC) (*Fang et al., 2017*). Here, we propose that CDPK4-dependent phosphorylation of SOC2 and/or CRK5 activates the kinase activity to phosphorylate components of the pre-replicative complex allowing initiation of DNA replication. Indirect regulation of dyneins through phosphatases such as PPM11 could possibly regulate progression into M-phase. Yellow circles represent phosphorylation.

The online version of this article includes the following source data and figure supplement(s) for figure 4:

**Source data 1.** Panel A, CRK5-HA western blot.

*Figure 4 continued on next page*

*Figure 4 continued*

**Source data 2.** Panel B, SOC2-HA western blot.
**Source data 3.** The CSC complex is stable but dynamically phosphorylated during gametogony.
**Figure supplement 1.** Study of the CSC complex regulation.
**Figure supplement 1—source data 1.** Study of the CSC complex regulation.

*supplement 1F and G*) suggesting that SOC2-S5088/5089 phosphorylation is either not required or insufficient to regulate CSC complex function.

CRK5 is also phosphorylated dynamically during gametogony (*Invergo et al., 2017*). We show here that inhibition of CDPK4 with BKI-1294 affected the phosphorylation at positions S654 and S693 of immunoprecipitated CRK5 (*Figure 4D*). Interestingly, all eight detected phosphorylation sites in CRK5 are located in the 148-residue C-terminal extension (*Figure 4E*), which is only found in *Plasmodium* CRK5 (*Dorin-Semblat et al., 2013*). We considered that this extension may have an important role in the regulation of CRK5 activity, and to test this hypothesis we generated a parasite clonal line, CRK5$_{1-562}$, with the last 148 amino acids of CRK5 deleted, leaving an intact catalytic domain (*Figure 4—figure supplement 1H*). This CRK5 truncation produced the same phenotype as that of the CRK5-KO line, with a significant reduction in exflagellation and DNA replication (*Figure 4F and G*). These results suggest the *Plasmodium*-specific C-terminus extension of CRK5 is essential for its function and may act as a regulatory platform to coordinate cell cycle progression during gametogony.

## Discussion

In this study, we have identified an atypical cyclin-CDK pair that is essential for male gametogony in a malaria parasite. The divergent CDK-related kinase, CRK5, interacts with a *Plasmodium*-restricted cyclin, SOC2, and with another putative regulatory subunit to form a complex we have named CSC. We provide evidence of CRK5-dependent phosphorylation of multiple components of the pre-replicative complex at a conserved eukaryotic CDK motif. Altogether, this suggests a very early requirement for CRK5 during DNA replication. Further studies are necessary to link phosphorylation of these proteins with CRK5 activity and to characterise the functional relevance of their phosphorylation on the activity of the pre-replicative complex. Given our poor understanding of the mechanisms that regulate the rapid DNA synthesis and the three rounds of mitosis during gametogony, it is unclear whether components of the CSC complex are required for a single checkpoint at the onset of S-phase or for two independent checkpoints, necessary to enter S- and M-phases, respectively. In any case, the abundance of phosphorylated dyneins increased upon CRK5 deletion suggesting that CRK5-dependent regulation of M-phase would likely be indirect, possibly affecting microtubule-dependent processes such as formation or segregation of nuclear poles or mitotic spindle organisation (*Agircan et al., 2014*; *Raaijmakers and Medema, 2014*; *Figure 4H*).

CRK5, SOC2 and CDKrs have similar transcriptional profiles during asexual replication in erythrocytes with a peak in late trophozoites (*López-Barragán et al., 2011*), suggesting the CSC complex may have conserved control functions in cell cycle progression during schizogony. Consistent with this idea, CRK5 deletion reduced nuclear division during erythrocytic schizogony in *P. falciparum* (*Dorin-Semblat et al., 2013*), while SOC2 deletion led to slower proliferation of *P. berghei* in erythrocytes (*Fang et al., 2017*). Given the versatile nature of cyclin-CDK complex function in other organisms, it is possible that other CDKs or cyclins partially compensate for the loss of CRK5 or SOC2 in asexual blood stages. For example, CRK4 activity may compensate for the lack of CRK5 during schizogony, because it is also important for DNA replication during this developmental stage (*Ganter et al., 2017*).

Usually there are significant oscillations in the abundance of cyclins during the cell cycle. However, SOC2 shows no evidence of oscillation or changes in the level of its interaction with CRK5. Such changes may be undetectable in the experimental conditions and we cannot exclude more subtle changes that may regulate the CSC complex. The observed absence of cycling may also reflect the constraints imposed by the extremely rapid nature of gametogony. No marked Cyc1 oscillations were detected during the slower *P. falciparum* blood stage schizogony. Furthermore, the Cyc1/MRK complex was detected in mature segmented schizonts after Cyc1 is required to complete cytokinesis

(*Robbins et al., 2017*). Therefore, it is also possible that non-oscillating cell-cycle cyclins and stable cyclin/CDK complexes are a conserved feature of the various cycles of division within *Plasmodium* parasites. However, PbCyc3 showed fluctuating expression patterns during oocyst development (*Roques et al., 2015*). It thus remains unclear whether non-cycling cyclins and stable cyclin/CDK complexes represent conserved features of the various cell cycles of *Plasmodium* parasites.

We provide evidence that the C-terminal extension of CRK5 is important in the control of its function. Interestingly, SOC2 and the C-terminal extension of CRK5 are both Haemosporidia-specific. For example, they are absent in the related Apicomplexa parasite *Toxoplasma gondii* in which the homologue of CRK5 is also involved in DNA replication (*White and Suvorova, 2018*), highlighting a specific evolution of this cyclin/CDK system in *Plasmodium*. It is tempting to speculate that the CRK5 C-terminal extension and its phosphorylation are important in the regulation of CRK5 activity via conformational changes within the complex, but this hypothesis remains untested as the exact role of SOC2 in CRK5 regulation. Very few studied CDKs contain additional domains with known function. For example, phosphorylation of the Cdk16 N-terminal domain blocks binding to cyclin Y (*Mikolcevic et al., 2012*). Multiple CRK5 and SOC2 residues are phosphorylated in a CDPK4-dependent manner and it is possible that CDPK4 works as a *Plasmodium*-specific link to translate a calcium signal triggering gametogony to prime the CSC complex (*Figure 4H*). In addition, CDPK4-independent phosphorylation of CRK5 is also observed (*Invergo et al., 2017*) and other kinases or phosphatases may control CSC activity during the rounds of DNA replication and mitosis. However, given the absence of canonical checkpoint kinases (*Tewari et al., 2010*; *Solyakov et al., 2011*; *Brochet et al., 2015*), it is possible that the successive mitotic cycles of gametogony are separated from each other by a clock mechanism rather than by checkpoints that rely on signals from one event in the cell cycle to regulate the next.

*Plasmodium* species lack clear CDK orthologues (*Cao et al., 2014*) that are usually highly conserved across Eukarya and indispensable in yeasts and humans (*Malumbres, 2014*). This study indicates that CRK5, despite its divergence from canonical CDKs, possibly acts as a CDK by binding a divergent cyclin and phosphorylating classical CDK motifs on proteins that are targeted by conventional CDKs and involved in the initiation of DNA replication in human and yeast. However, it remains difficult to identify a specific direct functional orthologue due to functional similarities with multiple CDKs and even non-CDK cell-cycle kinases, and it is tempting to speculate that CRK5 and SOC2 cover the functional space of multiple cell-cycle regulators in other organisms, to satisfy the specificities of the *Plasmodium* cell cycle. Future functional analyses will have to confirm that CRK5 activity is regulated in a SOC2-dependent manner and the direct phosphorylation of the pre-replicative complex by CRK5.

# Materials and methods

### Key resources table

| Reagent type (species) or resource | Designation | Source or reference | Identifiers | Additional information |
|---|---|---|---|---|
| Strain, strain background (*Escherichia coli*) | BigEasy-TSA | Lucigen | Cat# 60224–1 | Electrocompetent cells |
| Transfected constructs (*P. berghei*) | PlasmoGEM vectors | https://plasmogem.sanger.ac.uk/ | Materials and method s section and supplementary figures | |
| Cell line (*P. berghei*) | ANKA 2.34 | *Billker et al., 2004* | | |
| Cell line (*P. berghei*) | 507cl1 | *Janse et al., 2006b* | | |
| Cell line (*P. berghei*) | 820cl1 | *Mair et al., 2010* | | |
| Cell line (*P. berghei*) | 615 | *Philip and Waters, 2015* | | |

*Continued on next page*

*Continued*

| Reagent type (species) or resource | Designation | Source or reference | Identifiers | Additional information |
|---|---|---|---|---|
| Cell line (*P. berghei*) | 507cl1 CRK5-KO | This study | | Available from the Nottingham laboratory |
| Cell line (*P. berghei*) | 615 CRK5-AID/HA | This study | | Available from the Geneva laboratory |
| Cell line (*P. berghei*) | 2.34 CRK5-GFP | This study | | Available from the Nottingham laboratory |
| Cell line (*P. berghei*) | 2.34 CRK5-3xHA | This study | | Available from the Geneva laboratory |
| Cell line (*P. berghei*) | 2.34 CDKrs-GFP | This study | | Available from the Nottingham laboratory |
| Cell line (*P. berghei*) | 820cl1 CDKrs-KO | This study | | Available from the Geneva laboratory |
| Cell line (*P. berghei*) | 820cl1 SOC2-KO | *Fang et al., 2017* | | Available from the Geneva laboratory |
| Cell line (*P. berghei*) | 820cl1 SOC2-3xHA | *Fang et al., 2017* | | Available from the Geneva laboratory |
| Cell line (*P. berghei*) | 820cl1 SOC2$^{S5088/89A}$-3xHA | This study | | Available from the Geneva laboratory |
| Cell line (*P. berghei*) | 820cl1 CRK5$_{1-562}$ | This study | | Available from the Geneva laboratory |
| Antibody | Anti-HA High Affinity from rat IgG1 (3F10) | Sigma | Cat# 0000011867423001 | IF(1:1000), WB (1:1000), IP (1:1000) |
| Antibody | Anti-c-Myc antibody produced in rabbit | Sigma | Cat# C3956 | WB (1:1000) |
| Antibody | Anti-α-Tubulin antibody from mouse (DM1A) | Sigma | Cat# T6199 | IF(1:1000), WB (1:1000) |
| Antibody | Goat anti-Rat IgG (H+L), Alexa Fluor 594 | Thermofisher | Cat# A-11007 | IF(1:1000) |
| Antibody | Goat anti-Mouse IgG (H+L), Alexa Fluor 488 | Thermofisher | Cat# A-11029 | IF(1:1000) |
| Antibody | Goat anti-Rabbit IgG (H+L), Alexa Fluor 488 | Thermofisher | Cat# A-11034 | IF(1:1000) |
| Sequence-based reagent | PCR primers | Microsynth | Materials and methods section and *Supplementary file 5* | |
| Chemical compound, drug | BKI-1294 | *Ojo et al., 2014* | | CDPK4 inhibitor (1 µM) |

## Ethics statement

The animal work performed in the United Kingdom passed an ethical review process and was approved by the UK Home Office. Work was carried out under UK Home Office Project Licenses (40/3344 and 30/3248) in accordance with the United Kingdom 'Animals (Scientific Procedures) Act 1986'. All animal experiments performed in Switzerland were conducted with the authorisation numbers GE/82/15 and GE/41/17, according to the guidelines and regulations issued by the Swiss Federal Veterinary Office.

## Parasite maintenance and transfection

*P. berghei* ANKA strain (*Vincke and Lips, 1948*)-derived clones 2.34 (*Billker et al., 2004*), 507cl1 (*Janse et al., 2006b*), 820cl1 (*Mair et al., 2010*), and 615 (*Solyakov et al., 2011*), together with derived transgenic lines, were grown and maintained in CD1 outbred mice. Six to ten week-old mice were obtained from Charles River laboratories, and females were used for all experiments. Mice were specific pathogen free (including mycoplasma pulmonis) and subjected to regular pathogen monitoring by sentinel screening. They were housed in individually ventilated cages furnished with a

cardboard mouse house and Nestlet, maintained at 21 ± 2°C under a 12 hr light/dark cycle, and given commercially prepared autoclaved dry rodent diet and water ad libitum. The parasitaemia of infected animals was determined by microscopy of methanol-fixed and Giemsa-stained thin blood smears.

For gametocyte production, parasites were grown in mice that had been phenyl hydrazine-treated three days before infection. One day after infection, sulfadiazine (20 mg/L) was added in the drinking water to eliminate asexually replicating parasites. Microgametocyte exflagellation was measured three or four days after infection by adding 4 μl of blood from a superficial tail vein to 70 μl exflagellation medium (RPMI 1640 containing 25 mM HEPES, 4 mM sodium bicarbonate, 5% fetal calf serum (FCS), 100 μM xanthurenic acid, pH 7.8). To calculate the number of exflagellation centres per 100 microgametocytes, the percentage of red blood cells (RBCs) infected with microgametocytes was assessed on Giemsa-stained smears. For gametocyte purification, parasites were harvested in suspended animation medium (SA; RPMI 1640 containing 25 mM HEPES, 5% FCS, 4 mM sodium bicarbonate, pH 7.20) and separated from uninfected erythrocytes on a Histodenz/Nycodenz cushion made from 48% of a Histodenz/Nycodenz stock (27.6% [w/v] Histodenz/Nycodenz [Sigma/Alere Technologies] in 5.0 mM TrisHCl, 3.0 mM KCl, 0.3 mM EDTA, pH 7.20) and 52% SA, final pH 7.2. Gametocytes were harvested from the interface. To induce CRK5-AID/HA degradation, 1 mM auxin dissolved in ethanol (0.2% final concentration) was added to purified gametocytes for one hour prior to activation by XA.

Schizonts for transfection were purified from overnight in vitro culture on a Histodenz cushion made from 55% of the Histodenz/Nycodenz stock and 45% PBS. Parasites were harvested from the interface and collected by centrifugation at 500 g for 3 min, resuspended in 25 μL Amaxa Basic Parasite Nucleofector solution (Lonza) and added to 10–20 μg DNA dissolved in 10 μl H$_2$O. Cells were electroporated using the FI-115 program of the Amaxa Nucleofector 4D. Transfected parasites were resupended in 200 μl fresh RBCs and injected intraperitoneally into mice. Parasite selection with 0.07 mg/mL pyrimethamine (Sigma) in the drinking water (pH ~4.5) was initiated one day after infection. Each mutant parasite was genotyped by PCR using three combinations of primers, specific for either the WT or the modified locus on both sides of the targeted region (experimental designs are shown in Figure supplements). For allelic replacements, sequences were confirmed by Sanger sequencing using the indicated primers. Controls using wild type DNA were included in each genotyping experiment; parasite lines were cloned when indicated.

## Generation of DNA targeting constructs

The oligonucleotides used to generate and genotype the mutant parasite lines are in **Supplementary file 5**.

### Restriction/ligation cloning

The C-termini of CRK5 and CDKrs were tagged with GFP by single crossover homologous recombination in the parasite. To generate these GFP lines, a region of the *crk5* or *cdkrs* gene downstream of the ATG start codon was amplified using primers T1851/T1852 and T2461/T2462, respectively, ligated into p277 vector, and transfected in ANKA line 2.34 as described previously (*Guttery et al., 2012*). Diagnostic PCRs were performed with primer 1 (IntT185 or intT246) and primer 2 (ol492) to confirm integration of the GFP targeting constructs. Schematic representations of the endogenous *crk5* and *cdkrs* loci, the constructs and the recombined loci are found in *Figure 1—figure supplement 1* and S3 respectively.

The gene-deletion targeting vector for *crk5* was constructed using the pBS-DHFR plasmid, which contains polylinker sites flanking a *T. gondii* dhfr/ts expression cassette conferring resistance to pyrimethamine, as described previously (*Tewari et al., 2010*). PCR primers N1001 and N1002 were used to generate a fragment of *crk5* 5′ upstream sequence from genomic DNA, which was inserted into ApaI and HindIII restriction sites upstream of the dhfr/ts cassette of pBS-DHFR. A fragment generated with primers N1003 and N1004 from the 3′ flanking region of *crk5* was then inserted downstream of the dhfr/ts cassette using EcoRI and XbaI restriction sites. The linear targeting sequence was released using ApaI/XbaI and the construct was transfected into the ANKA line 507cl1 expressing GFP. Diagnostic PCR was performed with primer 1 (IntN100) and primer 2 (ol248) to confirm integration of the targeting construct, and primer 3 (N100 KO1) and primer 4 (N100 KO2) were used

to confirm deletion of the *crk5* gene. A schematic representation of endogenous *crk5*, the constructs and the recombined locus is in *Figure 1—figure supplement 1*.

## PlasmoGEM vectors

3xHA and AID/HA tagging of CRK5, CDKrs-KO, CRK5$_{1-562}$ and SOC2 allelic replacement construct were generated using phage recombineering in *Escherichia coli* TSA strain with PlasmoGEM vectors (http://plasmogem.sanger.ac.uk/). For final targeting vectors not available in the PlasmoGEM repository, generation of knock-out and tagging constructs was performed using sequential recombineering and gateway steps as previously described (*Pfander et al., 2013*; *Pfander et al., 2011*). For each gene of interest (goi), the Zeocin-resistance/Phe-sensitivity cassette was introduced using oligonucleotides *goi* HA-F x *goi* HA-R and *goi* KO-F x *goi* KO-R for 3xHA tagging and KO targeting vectors, respectively. Insertion of the GW cassette following gateway reaction was confirmed using primer pairs GW1 x *goi* QCR1 and GW2 x *goi* QCR2. The modified library inserts were then released from the plasmid backbone using NotI. The CRK5-AID/HA targeting vector was transfected into the 615 parasite line (*Philip and Waters, 2015*), the CDKrs-KO, and CRK5$_{1-562}$ vectors into the 820cl1 line and the CRK5-3xHA vector into the 2.34 line.

Substitutions of SOC2$^{S5088/5089A}$ residues were introduced with a two-step strategy using λ Red-ET recombineering as described in *Brochet et al., 2014*. The first step involved the insertion by homologous recombination of a Zeocin-resistance/Phe-sensitivity cassette flanked by 5′ and 3′ sequences of the codon of interest, which is amplified using the *soc2*-delF x *soc2*-delR primer pair. Recombinant bacteria were then selected on Zeocin. The recombination event was confirmed by PCR and a second round of recombination replaced the Zeocin-resistance/Phe-sensitivity cassette with a PCR product containing the S5088/5089A substitutions amplified using *soc2*-mut-F with *soc2*-mutR primer pairs, respectively. Bacteria were selected on YEG-Cl kanamycin plates. Mutations were confirmed by sequencing vectors isolated from colonies sensitive to Zeocin with primers *soc2*-seqF to *soc2*-seqR. The modified library insert was then released from the plasmid backbone using NotI and the construct transfected into the 801cl1 line.

## Parasite phenotype analyses

Blood containing approximately 50,000 parasites of the CRK5-KO line was injected intraperitoneally into mice to initiate infections. Four to five days post infection, exflagellation and ookinete conversion were examined as described previously (*Guttery et al., 2012*) with a Zeiss AxioImager M2 microscope (Carl Zeiss, Inc) fitted with an AxioCam ICc1 digital camera. When indicated, gametocytes were pre-treated 5 min in SA supplemented with 1 µM BKI-1294 (*Ojo et al., 2014*) or 60 min in SA supplemented with 1 µM MG132. To analyse mosquito transmission, 30 to 50 *Anopheles stephensi* SD 500 mosquitoes were allowed to feed for 20 min on anaesthetised, infected mice whose asexual parasitaemia had reached 15% and were carrying comparable numbers of gametocytes as determined on Giemsa-stained blood films. To assess mid-gut infection, approximately 15 guts were dissected from mosquitoes on day 7, 14, and 21 post-feeding and oocysts were counted using the Zeiss AxioImager M2 microscope and a 63x oil immersion objective. Mosquito bite-backs were performed 21 days post-feeding using naive mice and blood smears were examined after 3–4 days.

## Immunofluorescence labelling

Gametocyte immunofluorescence assays were performed as previously described (*Volkmann et al., 2012*). For HA and α-tubulin staining, purified cells were fixed with 4% paraformaldehyde and 0.05% glutaraldehyde in PBS for 1 hr, permeabilised with 0.1% Triton X-100/PBS for 10 min and blocked with 2% BSA/PBS for 30 min. Primary antibodies were diluted in blocking solution (rat anti-HA clone 3F10, 1:1000; mouse anti-α-tubulin clone DM1A, 1:1000, both from Sigma-Aldrich). Anti-rat Alexa594, anti-mouse Alexa488, anti-rabbit Alexa 488, Anti-rabbit Alexa594 were used as secondary antibodies together with DAPI (all from Life technologies), all diluted 1:1000 in blocking solution. Confocal images were acquired with a LSM700 or a LSM800 scanning confocal microscope (Zeiss).

The CRK5-KO gametocytes were purified and activated in ookinete medium, then fixed at 15 min pa with 4% paraformaldehyde (PFA, Sigma) diluted in microtubule stabilising buffer (MTSB) for 10–15 min and added to poly-L-lysine coated slides. Immunocytochemistry was performed using primary mouse anti-α-tubulin mAb (Sigma-T9026; 1:1000) and secondary antibody Alexa 488 conjugated

anti-mouse IgG (Invitrogen-A11004; 1:1000). The slides were then mounted in Vectashield 19 with DAPI (Vector Labs) for fluorescence microscopy. Parasites were visualised on a Zeiss AxioImager M2 microscope fitted with an AxioCam ICc1 digital camera (Carl Zeiss, Inc).

## Flow cytometry analysis of gametocyte DNA content and CRK5/CDKrs-GFP fluorescence

DNA content of microgametocytes was determined by flow cytometry measurement of fluorescence intensity of cells stained with Vybrant dye cycle violet (life Technologies). Gametocytes were purified and resuspended in 100 µl of SA. Activation was induced by adding 100 µl of modified exflagellation medium (RPMI 1640 containing 25 mM HEPES, 4 mM sodium bicarbonate, 5% FCS, 200 µM xanthurenic acid, pH 7.8). To rapidly block gametogony, 800 µl of ice cold PBS was added and cells were stained for 30 min at 4˚C with Vybrant dye cycle violet and analysed using a Beckman Coulter Gallios 4. Microgametocytes were selected on fluorescence by gating on GFP positive microgametocytes when the 820cl1 parasite line or its derivatives were used. In this case, cell ploidy was expressed as a percentage of male gametocytes only. When the 2.34 line or its derivatives were analysed, gating was performed on both micro- and macro-gametocytes and cell ploidy was expressed as a percentage of all gametocytes. For each sample, >20,000 cells were analysed.

## Electron microscopy

Gametocytes samples at 1, 3, 6, 15 and 30 min pa were fixed in 4% glutaraldehyde in 0.1 M phosphate buffer. Samples were post fixed in osmium tetroxide, treated *en bloc* with uranyl acetate, dehydrated and embedded in Spurr's epoxy resin. Thin sections were stained with uranyl acetate and lead citrate prior to examination in a JEOL1200EX electron microscope (Jeol UK Ltd).

## Phylogenetic analyses

To search for orthologs of *Plasmodium* CRK sequences in alveolates, annotated protein sequences were downloaded for the following organisms: *P. falciparum* (pfal), *P. reichenowi* (prei), *P. knowlesi* (pkno), *P. vivax* (pviv), *P. berghei* (pber), *P. chabaudi* (pcha), and *P. yoelii* (all from PlasmoDB, [plasmodb.org]). *Theileria annulata* (tann), *T. parva* (tpar), and *Babesia bovis* (bbov) (all from PiroplasmaDB, [piroplasmadb.org]). *Toxoplasma gondii* (tgon), *Neospora caninum* (ncan), and *Eimeria tenella* (eten) (all from ToxoDB, [toxodb.org]). *Cryptosporidium hominis* (chom), *C. parvum* (cpar), *Chromera velia* (cvel), and *Vitrella brassicaformis* (vbra) (all from CryptoDB, [cryptodb.org]). *Perkinsus marinus* (pmar), *Ichthyophthirius multifiliis* (imul), and *Tetrahymena thermophilia* (tthe) (all from EnsemblProtists, [protists.ensembl.org]). *Symbiodinium minutum* (smin) (from OIST, [marinegenomics.oist.jp]).

Reciprocal best hits to *P. berghei* CRK sequences were determined from pair-wise protein BLAST searches. For each CRK protein, reciprocal best BLAST matches were retrieved and aligned using MAFFT (*Katoh and Standley, 2013*). Alignments were trimmed using trimal (*Capella-Gutiérrez et al., 2009*) and HMM models were produced with HMMer (hmmer.org). These models were then used to search the genomes for which no reciprocal best BLAST hit was detected. A neighbour-joining tree was then constructed using four combined data sets: 1) the *P. berghei* CRK proteins, 2) all protein sequences with a reciprocal best BLAST hit to *P. berghei*, 3) genes that were best BLAST hit for a *P. berghei* protein, but did not have the CRK protein as best hit in *P. berghei*, 4) all protein sequences with a score below $1 \times 10^{-12}$ to an HMM model. Sequences and identifiers are provided in the source data files of *Figure 3—figure supplement 1*.

Sequence distances were calculated using clustalw2 (*Larkin et al., 2007*; *Thompson et al., 1994*), which was also used to construct the NJ tree. Protein sequences from 1) and 2) were divided into their CRK groups (*i.e.* CRK1-7, CDKrs, and SOC2), and all pair-wise distances were calculated within each CRK group. For all protein sequences from 3) and 4), the distance to all members in all CRK groups were then measured. A protein sequence was included in a CRK group if the average distance between the protein sequence and all members of a given CRK group did not exceed 40% of all pair-wise distances within that CRK group, or if the CRK group to which a protein sequence had the lowest average distance was lower than the distance to the CRK group with the second-lowest average distance minus five standard deviations. Final trees were produced by PhyML (*Guindon et al., 2010*; *Guindon and Gascuel, 2003*) with the GTR substitution model selected by

SMS (*Lefort et al., 2017*). Branch support was evaluated with the Bayesian-like transformation of approximate likelihood ratio test, aBayes (*Anisimova et al., 2011*). Trees were visualised using Fig-Tree (tree.bio.ed.ac.uk/software/figtree/).

## Transcriptomics

Gametocytes (non-activated −0 min- and activated −15 min-) were collected from CRK5-KO and WT lines. Total RNA was isolated from purified parasites using an RNeasy purification kit (Qiagen). RNA was vacuum concentrated (SpeedVac) and transported using RNA stable tubes (Biomatrica). Strand-specific mRNA sequencing was performed from total RNA using TruSeq Stranded mRNA Sample Prep Kit LT (Illumina) according to the manufacturer's instructions. Briefly, polyA+ mRNA was puri-fied from total RNA using oligo-dT dynabead selection. First strand cDNA was synthesised primed with random oligos followed by second strand synthesis where dUTPs were incorporated to achieve strand-specificity. The cDNA was adapter-ligated and the libraries amplified by PCR. Libraries were sequenced in an Illumina Hiseq machine with paired-end 100 bp read chemistry.

RNA-seq read alignments were mapped onto the *P. berghei* ANKA genome (May 2015 release in GeneDB—http://www.genedb.org/) using Tophat2 (version 2.0.13) with parameters '-library-type fr-firststrand–no-novel-juncs–r 60'. Transcript abundances were extracted as raw read counts using the Python script 'HTseq-count' (*Anders et al., 2015*) (model type – union, http://www-huber.embl.de/users/anders/HTSeq/). Counts per million (cpm) values were obtained from count data and genes were filtered if they failed to achieve a cpm value of 1 in at least 30% of samples per condition. Library sizes were normalised by the TMM method using EdgeR software (*McCarthy et al., 2012*) and further subjected to linear model analysis using the voom function in the limma package (*Ritchie et al., 2015*). Differential expression analysis was performed using DESeq2 in R version 3.2.1 (*Love et al., 2014*). Genes with fold change greater than two and p-value less than 0.05 were considered as significantly differentially expressed. *P. berghei* GO terms (Gene Ontology) were downloaded from GeneDB (http://www.genedb.org/; May 2015 release) and gene ontology enrich-ment analysis was performed for the DEG lists using GOstats R package. All analyses and visualisa-tions were done with R packages- *cummeRbund* and *ggplot2.*

## Quantitative real time PCR (qRT-PCR) analyses

RNA was isolated from purified gametocytes using an RNA purification kit (Stratagene). cDNA was synthesised using an RNA-to-cDNA kit (Applied Biosystems). Gene expression was quantified from 80 ng of total RNA using SYBR green fast master mix kit (Applied Biosystems). All the primers were designed using primer3 (Primer-blast, NCBI). Analysis was conducted using an Applied Biosystems 7500 fast machine with the following cycling conditions: 95 °C for 20 s followed by 40 cycles of 95 °C for 3 s; 60 °C for 30 s. Three technical replicates and two biological replicates were performed for each assayed gene. The *hsp70* (PBANKA_081890) and *arginyl-t RNA synthetase* (PBANKA_143420) genes were used as endogenous control reference genes. The primers used for qPCR can be found in *Supplementary file 5*.

## Protein immunoprecipitation and identification

### Sample preparation

Co-immunoprecipitations (IPs) of proteins were performed with purified gametocytes. The following IPs were performed: SOC2-3xHA (0, 15, 30 and 60 sec pa), CDKrs-GFP (60 s, 6 min and 30 min pa), CRK5-3xHA (0, 15, 30 and 60 sec pa), and CRK5-GFP (at 6 and 15 min pa). IP from wild type non-activated gametocytes lacking an epitope tag or constitutively expressing GFP were used as controls.

For HA-based IPs, samples were fixed for 10 min with 1% formaldehyde. Parasites were lysed in RIPA buffer (50 mM Tris HCl pH 8, 150 mM NaCl, 1% NP-40, 0.5% sodium deoxycholate, 0.1% SDS) and the supernatant was subjected to affinity purification with anti-HA antibody (Sigma) or anti-GFP antibody (Invitrogen) conjugated to magnetics beads. Beads were re-suspended in 100 µl of 6 M urea in 50 mM ammonium bicarbonate (AB). Two µl of 50 mM dithioerythritol (DTE) were added and the reduction was carried out at 37°C for 1 hr. Alkylation was performed by adding 2 µl of 400 mM iodoacetamide for 1 hr at room temperature in the dark. Urea was reduced to 1 M by addition of 500 µl AB and overnight digestion was performed at 37°C with 5 µl of freshly prepared 0.2 µg/µl

trypsin (Promega) in AB. Supernatants were collected and completely dried under speed-vacuum. Samples were then desalted with a C18 microspin column (Harvard Apparatus) according to manufacturer's instructions, completely dried under speed-vacuum and stored at −20°C.

## Liquid chromatography electrospray ionisation tandem mass spectrometry (LC-ESI-MSMS)

Samples were diluted in 20 µl loading buffer (5% acetonitrile [$CH_3CN$], 0.1% formic acid [FA]) and 2 µl were injected onto the column. LC-ESI-MS/MS was performed either on a Q-Exactive Plus Hybrid Quadrupole-Orbitrap Mass Spectrometer (Thermo Fisher Scientific) equipped with an Easy nLC 1000 liquid chromatography system (Thermo Fisher Scientific) or an Orbitrap Fusion Lumos Tribrid mass Spectrometer (Thermo Fisher Scientific) equipped with an Easy nLC 1200 liquid chromatography system (Thermo Fisher Scientific). Peptides were trapped on an Acclaim pepmap100, 3 µm C18, 75 µm x 20 mm nano trap-column (Thermo Fisher Scientific) and separated on a 75 µm x 250 mm (Q-Exactive) or 500 mm (Orbitrap Fusion Lumos), 2 µm C18, 100 Å Easy-Spray column (Thermo Fisher Scientific). The analytical separation used a gradient of $H_2O$/0.1% FA (solvent A) and $CH_3CN$/0.1% FA (solvent B). The gradient was run as follows: 0 to 5 min 95% A and 5% B, then to 65% A and 35% B for 60 min, then to 10% A and 90% B for 10 min and finally for 15 min at 10% A and 90% B. Flow rate was 250 nL/min for a total run time of 90 min.

Data-dependant analysis (DDA) was performed on the Q-Exactive Plus with MS1 full scan at a resolution of 70,000 Full width at half maximum (FWHM) followed by MS2 scans on up to 15 selected precursors. MS1 was performed with an AGC target of $3 \times 10^6$, a maximum injection time of 100 ms and a scan range from 400 to 2000 m/z. MS2 was performed at a resolution of 17,500 FWHM with an automatic gain control (AGC) target at $1 \times 10^5$ and a maximum injection time of 50 ms. Isolation window was set at 1.6 m/z and 27% normalised collision energy was used for higher-energy collisional dissociation (HCD). DDA was performed on the Orbitrap Fusion Lumos with MS1 full scan at a resolution of 120,000 FWHM followed by as many subsequent MS2 scans on selected precursors as possible within a 3 s maximum cycle time. MS1 was performed in the Orbitrap with an AGC target of $4 \times 10^5$, a maximum injection time of 50 ms and a scan range from 400 to 2000 m/z. MS2 was performed in the Ion Trap with a rapid scan rate, an AGC target of $1 \times 10^4$ and a maximum injection time of 35 ms. Isolation window was set at 1.2 m/z and 30% normalised collision energy was used for HCD.

## Database searches

Peak lists (MGF file format) were generated from raw data using the MS Convert conversion tool from ProteoWizard. The peak list files were searched against the PlasmoDB_*P.berghei* ANKA database (PlasmoDB.org, release 38, 5076 entries) combined with an in-house database of common contaminants using Mascot (Matrix Science, London, UK; version 2.5.1). Trypsin was selected as the enzyme, with one potential missed cleavage. Precursor ion tolerance was set to 10 ppm and fragment ion tolerance to 0.02 Da for Q-Exactive Plus data and to 0.6 for Lumos data. Variable amino acid modifications were oxidised methionine and deamination (Asn and Gln) as well as phosphorylated serine, threonine and tyrosine. Fixed amino acid modification was carbamidomethyl cysteine. The Mascot search was validated using Scaffold 4.8.4 (Proteome Software) with 1% of protein false discovery rate (FDR) and at least two unique peptides per protein with a 0.1% peptide FDR.

## PCA analysis

Enrichment and principal component analysis was performed in the statistical programming package 'R' (www.r-project.org). Quantitative values were analysed as log-transformed spectral count values and displayed in principal components with greatest degrees of variance.

## Label-free comparison of peptide and phosphopeptides in SOC2 or CRK5 immunoprecipitates

Label-free comparison of peptide and phosphopeptides in SOC2 or CRK5 immunoprecipitates during the first minute of gametogony was determined using Proteome Discoverer 2.2 (Thermo Fisher Scientific). Peptide-spectrum matches were validated using Percolator validator node with a target FDR of 0.01 and a Delta Cn of 0.5. For label-free quantification, features and chromatographic peaks

were detected using the 'Minora Feature Detector' Node with the default parameters. PSM and peptides were filtered with a false discovery rate (FDR) of 1%, and then grouped to proteins with again a FDR of 1% and using only peptides with high confidence level. Both unique and razor peptides were used for quantitation and protein abundances are calculated by summing samples abundances of the connected peptides group. The abundances were normalised on the 'Total Peptide Amount' and then 'The pairwise Ratio Based' option was selected for protein ratio calculation and associated p-values were calculated with an ANOVA test based on background.

## Proteomics and phosphoproteomics

### Sample preparation (SDS buffer-FASP procedure)

Activated and non-activated purified gametocytes were snap frozen in liquid nitrogen at 0, 15, 30, 45 and 60 sec pa. For each time point, two independent biological replicates were analysed. Cell lysis was performed in 1 ml of 2% SDS, 25 mM NaCl, 2.5 mM EDTA, 20 mM TCEP, 50 mM TrisHCl (pH 7.4), supplemented with 1x Halt protease and phosphatase inhibitor (Thermo Fisher). Samples were vortexed and then heated at 95°C for 10 min with mixing at 400 rpm on a thermomixer. DNA was sheared *via* four sonication pulses of 10 s each at 50% power. Samples were then centrifuged for 30 min at 17,000 $g$ and the supernatant was collected. Three-hundred μl sample was incubated with 48 μl 0.5 M iodoacetamide for 1 hr at room temperature. Protein was digested based on the FASP method (*Wiśniewski et al., 2009*) using Amicon Ultra-4, 30 kDa cut-off as centrifugal filter units (Millipore). Trypsin (Promega) was added at 1:100 enzyme/protein ratio and digestion was performed at 37°C overnight. The resulting peptides were treated as described above.

### TMT11plex-labelling

Peptide concentration was determined using a colorimetric peptide assay (Pierce). A pool made from 1/20 of each sample was created and used as reference for each tandem mass tag (TMT) experiment. Briefly, 100 μg of each sample was labelled with 400 μg of corresponding TMT-11plex reagent previously dissolved in 110 μl 36% $CH_3CN$, 200 mM 4-(2-hydroxyethyl)−1-piperazinepropanesulfonic acid (EPPS, pH 8.5). The reference sample was labelled with TMT11-131C label reagent. The reaction was performed for 1 hr at room temperature and then quenched by adding hydroxylamine to a final concentration of 0.3% (v/v). Labelled samples of each TMT11 experiment were pooled together, dried and desalted with a peptide desalting spin column (Pierce) according to manufacturer's instructions.

### Phosphopeptide enrichment

Phosphopeptides were enriched using the High-Select Fe-NTA Phosphopeptide Enrichment Kit (Thermo Fisher Scientific) following manufacturer's instructions. The phosphopeptide fraction as well as the flow-through fraction were desalted with a C18 macrospin column (Harvard Apparatus) according to manufacturer's instruction and then completely dried under speed-vacuum.

### High pH Reverse-Phase fractionation

The flow-through fraction of each TMT11 experiment was fractionated into 13 fractions using the Pierce High pH Reversed-Phase Peptide Fractionation Kit (Thermo Fisher Scientific) according to manufacturer's instructions.

### LC-ESI-MSMS

For all samples, peptide concentration was determined using a colorimetric peptide assay (Pierce). Phosphopeptides were reconstituted in loading buffer (5% $CH_3CN$, 0.1% FA) and 2 μg injected on to the column. LC-ESI-MS/MS was performed on an Orbitrap Fusion Lumos Tribrid mass spectrometer (Thermo Fisher Scientific) equipped with an Easy nLC1200 liquid chromatography system (Thermo Fisher Scientific). Peptides were trapped on a Acclaim pepmap100, C18, 3 μm, 75 μm x 20 mm nano trap-column (Thermo Fisher Scientific) and separated on a 75 μm x 500 mm, C18, 2 μm, 100 Å Easy-Spray column (Thermo Fisher Scientific). The analytical separation was run for 125 min using a gradient of 99.9% $H_2O$/0.1% FA (solvent A) and 80% $CH_3CN$/0.1% FA (solvent B). The gradient was run as follows: 0 to 2 min 92% A and 8% B, then to 72% A and 28% B for 105 min, to 58% A and 42% B for 20 min, and finally to 5% A and 95% B for 10 min with a final 23 min at this

composition. Flow rate was of 250 nL/min. DDA was performed with MS1 full scan at a resolution of 120,000 FWHM followed by as many subsequent MS2 scans on selected precursors as possible within a 3 s maximum cycle time. MS1 was performed in the Orbitrap with an AGC target of $4 \times 10^5$, a maximum injection time of 50 ms and a scan range from 375 to 1500 $m/z$. MS2 was performed in the Orbitrap using HCD at 38% Normalised collision Energy (NCE). Isolation window was set at 0.7 u with an AGC target of $5 \times 10^4$ and a maximum injection time of 86 ms. A dynamic exclusion of parent ions of 60 s with 10 ppm mass tolerance was applied.

High pH Reversed-Phase Peptide fractions were reconstituted in the loading buffer (5% $CH_3CN$, 0.1% FA) and 1 µg was injected onto the column. LC-ESI-MS/MS was performed as described above. The analytical separation was run for 90 min using a gradient of 99.9% $H_2O$/0.1% FA (solvent A) and 80% $CH_3CN$/0.1%FA (solvent B). The gradient was run as follows: 0 to 5 min 95% A and 5% B, then to 65% A and 35% B for 60 min, and finally to 5% A and 95% B for 10 min with a final 15 min at this composition. Flow rate was 250 nL/min. DDA was performed using the same parameters, as described above.

### Database search

Raw data were processed using Proteome Discoverer 2.3 software (Thermo Fisher Scientific). Briefly, spectra were extracted and searched against the *P. berghei* ANKA database (PlasmoDB.org, release 38, 5076 entries) combined with an in-house database of common contaminants using Mascot (Matrix Science, London, UK; version 2.5.1). Trypsin was selected as the enzyme, with one potential missed cleavage. Precursor ion tolerance was set to 10 ppm and fragment ion tolerance to 0.02 Da. Carbamidomethyl of cysteine (+57.021) as well as TMT10plex (+229.163) on lysine residues and on peptide N-termini were specified as fixed modifications. Oxidation of methionine (+15.995) as well as phosphorylated serine, threonine and tyrosine were set as variable modifications. The search results were validated with a Target Decoy PSM validator. PSM and peptides were filtered with a FDR of 1%, and then grouped to proteins, again with a FDR of 1% and using only peptides with high confidence level. Both unique and razor peptides were used for quantitation and protein and peptides abundances value were based on signal to noise (S/N) values of reporter ions. Abundances were normalised on 'Total Peptide Amount' and then scaled with 'On Controls Average' (i.e. using the reference sample channel). All the protein ratios were calculated from the medians of the summed abundances of replicate groups and associated p-values were calculated with an ANOVA test based on individual protein or peptides. GO term enrichment was performed using PlasmoDB. The reference set of GO terms for human kinases was obtained from *Ganter et al., 2017*.

## Statistical analysis

All statistical analyses were performed using GraphPad Prism 8 (GraphPad Software) assuming normal distribution. Statistical tests are mentioned in the figure legends.

## Acknowledgements

We thank Julie Rodger (Nottingham University) for her assistance in the insectary maintenance and Zineb Rchiad (KAUST) for RNAseq library preparation. We thank the excellent service at the bioimaging and flow-cytometry core facilities at the Faculty of Medicine of the University of Geneva. We also would like to thank Nisha Philip (University of Edinburgh) for sharing the 615 Tir1-expressing line as well as Wesley Van Voorhis and Kayode Ojo (University of Washington) for sharing compound BKI-1294. We thank Markus Ganter for sharing the reference GO term set used in this work.

This work was supported by the Swiss National Science Foundation grant BSSGI0_155852 and 31003A_179321 to MB. MB is an INSERM and EMBO young investigator. The work in RT lab is supported by Medical Research Council UK (G0900109, G0900278, MR/K011782/1) and Biotechnology and Biological Sciences Research Council (BB/N017609/1). The work in AP lab is supported by a faculty baseline fund (BAS/1/1020-01-01) and a Competitive Research Grant (CRG) award from OSR (OSR-2018-CRG6-3392) from the King Abdullah University of Science and Technology. AAH was supported by the Francis Crick Institute (FC010097), which receives its core funding from Cancer Research UK (FC010097), the UK Medical Research Council (FC010097), and the Wellcome Trust (FC010097).

## Additional information

### Funding

| Funder | Grant reference number | Author |
|---|---|---|
| Schweizerischer Nationalfonds zur Förderung der Wissenschaftlichen Forschung | BSSGI0_155852 | Mathieu Brochet |
| Schweizerischer Nationalfonds zur Förderung der Wissenschaftlichen Forschung | 31003A_179321 | Mathieu Brochet |
| European Molecular Biology Organization | | Mathieu Brochet |
| Medical Research Council | G0900109 | Rita Tewari |
| Medical Research Council | G0900278 | Rita Tewari |
| Medical Research Council | MR/K011782/1 | Rita Tewari |
| Biotechnology and Biological Sciences Research Council | BB/N017609/1 | Rita Tewari |
| Francis Crick Institute | FC010097 | Anthony A Holder |
| Cancer Research UK | FC010097 | Anthony A Holder |
| Medical Research Council | FC010097 | Anthony A Holder |
| Wellcome Trust | FC010097 | Anthony A Holder |
| King Abdullah University of Science and Technology | OSR-2018-CRG6-3392 | Arnab Pain |
| King Abdullah University of Science and Technology | BAS/1/1020-01-01 | Arnab Pain |

The funders had no role in study design, data collection and interpretation, or the decision to submit the work for publication.

### Author contributions

Aurélia C Balestra, Conceptualization, Formal analysis, Validation, Investigation, Visualization, Methodology, Writing - review and editing; Mohammad Zeeshan, Conceptualization, Formal analysis, Validation, Investigation, Visualization, Methodology; Edward Rea, Patrizia Arboit, Rajan Pandey, Declan Brady, Investigation; Carla Pasquarello, Natacha Klages, David JP Ferguson, Formal analysis, Investigation, Methodology; Lorenzo Brusini, Formal analysis, Investigation, Visualization, Methodology; Tobias Mourier, Amit Kumar Subudhi, Formal analysis; Sue Vaughan, Resources; Anthony A Holder, Conceptualization, Writing - review and editing; Arnab Pain, Funding acquisition; Alexandre Hainard, Formal analysis, Supervision, Investigation, Methodology; Rita Tewari, Conceptualization, Resources, Formal analysis, Supervision, Funding acquisition, Investigation, Methodology, Project administration, Writing - review and editing; Mathieu Brochet, Conceptualization, Resources, Formal analysis, Supervision, Funding acquisition, Investigation, Visualization, Methodology, Writing - original draft, Project administration, Writing - review and editing

### Author ORCIDs

Anthony A Holder  http://orcid.org/0000-0002-8490-6058
David JP Ferguson  http://orcid.org/0000-0001-5045-819X
Rita Tewari  https://orcid.org/0000-0003-3943-1847
Mathieu Brochet  https://orcid.org/0000-0003-3911-5537

### Ethics

Animal experimentation: The animal work performed in the United Kingdom passed an ethical review process and was approved by the UK Home Office. Work was carried out under UK Home Office Project Licenses (40/3344 and 30/3248) in accordance with the United Kingdom 'Animals

(Scientific Procedures) Act 1986'. All animal experiments performed in Switzerland were conducted with the authorisation numbers GE/82/15 and GE/41/17, according to the guidelines and regulations issued by the Swiss Federal Veterinary Office.

### Decision letter and Author response
Decision letter https://doi.org/10.7554/eLife.56474.sa1
Author response https://doi.org/10.7554/eLife.56474.sa2

## Additional files

### Supplementary files
- Supplementary file 1. RNAseq data.
- Supplementary file 2. Peptides detected in WT and CRK5-KO gametocytes.
- Supplementary file 3. Phosphopeptides detected in WT and CRK5-KO gametocytes.
- Supplementary file 4. Spectral count values of proteins identified in co-immunoprecipitates by mass-spectrometry.
- Supplementary file 5. Oligonucleotides used in this study.
- Supplementary file 6. Correspondence between TMT labels and samples.
- Transparent reporting form

### Data availability
Mass spectrometry proteomics data have been deposited to the ProteomeXchange Consortium via the PRIDE partner repository with the dataset identifier PXD017283 (proteomics and phosphoproteomics), PXD017308 (HA immunoprecipitations) and PXD017622 (GFP immunoprecipitations). RNA-seq data have been deposited to the Gene Expression Omnibus under accession number GSE144743. The authors declare that all other relevant data generated or analysed during this study are included in the article or its supplementary information. Raw data are available for each figure. Materials are available from the corresponding authors on reasonable request.

The following datasets were generated:

| Author(s) | Year | Dataset title | Dataset URL | Database and Identifier |
|---|---|---|---|---|
| Hainard A | 2020 | A divergent cyclin/cyclin-dependent kinase complex controls progression through the atypical replicative cycles during Plasmodium berghei gametogony | http://www.ebi.ac.uk/pride/archive/projects/PXD017308 | PRIDE, PXD017308 |
| Hainard A | 2020 | A divergent cyclin/cyclin-dependent kinase complex controls progression through the atypical replicative cycles during Plasmodium berghei gametogony | http://www.ebi.ac.uk/pride/archive/projects/PXD017283 | PRIDE, PXD017283 |
| Bottrill A | 2020 | A divergent cyclin/cyclin-dependent kinase complex controls the atypical replication of Plasmodium berghei during gametogony and parasite transmission | http://www.ebi.ac.uk/pride/archive/projects/PXD017622 | PRIDE, PXD017622 |
| Balestra A, Zeeshan M, Subudhi Kumar A, Pain A, Tewari R, Brochet M | 2020 | A divergent cyclin/cyclin-dependent kinase complex controls the atypical replication of Plasmodium berghei during gametogony and parasite transmission | https://www.ncbi.nlm.nih.gov/geo/query/acc.cgi?acc=GSE144743 | NCBI Gene Expression Omnibus, GSE144743 |

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
