## [Decision Letter]

**Acceptance summary:**

This paper uncovers an important protein complex at the center of the unusual cell cycle of malaria parasites (*Plasmodium spp*.). Displaying distant homology to cyclins and cyclin-dependent protein kinases, these proteins display a critical function in the regulation of the incredibly rapid cell divisions that form male gametocytes once parasites reach the mosquito. The advance will contribute to an improved understanding of cell-cycle regulation in these medically important pathogens.

**Decision letter after peer review:**

Thank you for submitting your article "A divergent cyclin/cyclin-dependent kinase complex controls replication of a malaria parasite during transmission" for consideration by *eLife*. Your article has been reviewed by three peer reviewers, one of whom is a member of our Board of Reviewing Editors, and the evaluation has been overseen by Philip Cole as the Senior Editor. The following individuals involved in review of your submission have agreed to reveal their identity: Elena Suvorova (Reviewer #2); Moritz Treeck (Reviewer #3).

The reviewers have discussed the reviews with one another and the Reviewing Editor has drafted this decision to help you prepare a revised submission.

As the editors have judged that your manuscript is of interest, but as described below that additional experiments are required before it is published, we would like to draw your attention to changes in our revision policy that we have made in response to COVID-19 (https://elifesciences.org/articles/57162). First, because many researchers have temporarily lost access to the labs, we will give authors as much time as they need to submit revised manuscripts. We are also offering, if you choose, to post the manuscript to bioRxiv (if it is not already there) along with this decision letter and a formal designation that the manuscript is “in revision at *eLife*”. Please let us know if you would like to pursue this option. (If your work is more suitable for medRxiv, you will need to post the preprint yourself, as the mechanisms for us to do so are still in development.)

Summary:

Progression through gametogony is critical for the transmission cycle of malaria parasites. A key feature of this progression is the development of male gametocytes, which undergo three rounds of record-speed DNA replication and segregation following activation in the mosquito midgut, through a process termed exflagellation. However, a lack of canonical cyclins and cyclin-dependent protein kinases (CDKs) leaves open critical questions about the regulation of cell cycle progression through the complex life cycle of malaria parasites and related apicomplexans.

Balestra and co-authors describe the identification of an atypical CDK-cyclin complex that plays a major role regulating male gametogenesis in malaria parasites. The authors propose that a cyclin-related kinase (CRK5) is part of a stable complex with an unusual cyclin-like protein (SOC2), and a CDK regulatory subunit (CDKrs). Using quantitative phosphoproteomics the authors show that CRK5 loss affects the phosphorylation of components of the pre-replicative complex and the mitotic spindle, consistent with profound defects in endomitosis. Investigating the regulation of the complex, the authors demonstrate that, unlike canonical CDK-cyclin complexes, CRK5, SOC2, and CDKrs remain stably associated. The data instead suggest that C-terminal phosphorylation of CRK5 by CDPK4 acts as an alternative mechanism of activation.

Genetic evidence for the complex and its critical role in cell-cycle progression during gametogony is convincing, and an important advance for the field. However, the divergent cyclin/CDK complex should be supported through more careful analysis and evidence of cyclin-dependent activity. Some of the claims attributing molecular function to the complex, also need to be more carefully written, absent further experimental evidence.

Essential revisions:

The major concerns of the reviewers fall into three general categories:

1) Complex formation and mass spectrometry.

a) The mass-spectrometry experiments are poorly explained and the tables poorly annotated. There is no mention in the text of where TMT labelling was used, numbers of replicates, numbers of proteins identified etc. Fold-change significance should ideally be based on median standard deviation of the dataset, not an arbitrary cut-off. Little information is provided for the IP datasets, which are the foundation of the CSC complex and it seems critical to include control IPs against parasites of similar stages lacking an epitope tag, to establish the background of non-specific binding.

b) IPs for CRK5, SOC2, and CDKrs all recovered the other putative complex components, but also several members of the MCM and ORC complexes. The authors should more precisely define the complex arrangement, since CDK-cyclin complexes canonically comprise two components despite their numerous associations throughout the cell. An experiment establishing the stoichiometry of components, or their hierarchy of association should aid in defining the putative complex. As described below, linking the complex composition to its function through a more detailed analysis of the CRK5 C-terminus could fulfil concerns about the complex, as well as about its activity.

2) Complex activity.

The authors should establish that the complex is either activated during gametogony or dependent on SOC2 for activity. In vitro kinase assays could be performed as described by Dorin-Semblat et al., 2013. Alternatively, the activity of complexes isolated from the WT and SOC2-defficient parasites, or before and after activation, could help establish the conditional activation of the complex.

3) Complex Function.

There is insufficient evidence to establish the CRK5 control of both S and M phases (Abstract) and the authors should tone down their conclusions regarding the precise function of the complex. It is expected that blocking DNA replication will not wholly block preparation for mitosis. Concurrent events are likely controlled in parallel or partially independent ways. Furthermore, CRK4 is involved in S/M phase progression at the RBC stage and may also play a role in gametogenesis (Ganter et al., 2017); authors consider this possibility (Discussion paragraph two) but do not include it in their final statements (Abstract). Furthermore, changes in phosphorylation are not sufficient to establish regulation, especially since the possibility that the block in cell cycle progression has not caused a change in total protein abundance.

[Editors' note: further revisions were suggested prior to acceptance, as described below.]

Thank you for re-submitting your article "A divergent cyclin/cyclin-dependent kinase complex controls replication of a malaria parasite during transmission" for consideration by *eLife*. Your article was re-reviewed by the three peer reviewers including Sebastian Lourido as the Reviewing Editor and Reviewer #1, and the evaluation has been overseen by Philip Cole as the Senior Editor. The following individuals involved in review of your submission have agreed to reveal their identity: Elena Suvorova (Reviewer #2); Moritz Treeck (Reviewer #3).

The reviewers have discussed the reviews with one another and the Reviewing Editor has drafted this decision to help you prepare a revised submission.

We would like to draw your attention to changes in our revision policy that we have made in response to COVID-19 (https://elifesciences.org/articles/57162).

Summary:

Progression through the cell cycle is governed by divergent cell cycle kinases in Apicomplexan parasites. The authors characterize the role of one such kinase, CRK5, during the incredibly fast replicative cycle of the male gametocytes, following activation of the precursor in the mosquito midgut. The manuscript makes two claims of significant importance: that CRK5 functions as part of a complex (CSC) that includes proteins that are distantly homologous to CDK regulatory subunits, and that the CSC complex regulates key cell cycle transitions during exflagellation. The authors also provide intriguing evidence that suggests regulation of this complex by CDPK4 integrating the function of the complex into a broader regulatory network.

The revised manuscript addresses many of the issues raised during the previous round of reviews, and provide more convincing support for the constitution of the CSC. However, the reviewers all agree that three main issues require additional clarification prior to acceptance for publication. This therefore represents a conditional acceptance, providing the authors faithfully address the reviewer's comments.

Essential revisions:

1) Proteomics: Authors should use the same gene IDs in the source data files as in the text. All columns should be labelled and a legend should explain how the values for that column were derived. Log_2_FCs for each phosphorylation site should be clearly visible with most important datapoints (localisation score, quality of MS assignment, peptide identified) for each timepoint. The authors should comment and discuss why some of the proteins are already differentially phosphorylated at t=0 (for example ORC2).

2) CDK nomenclature and function: The authors are meticulous throughout the text stating that their conclusions are suggestive of CDK function, but the final statements of the Introduction or Discussion are not supported by the results e.g. "This study indicates that CRK5, despite its divergence from canonical CDKs, acts as a genuine CDK by binding a cyclin and phosphorylating classical CDK motifs on proteins that are targeted by conventional CDKs and involved in the initiation of DNA replication in human and yeast." The reviewers instruct the authors to limit their claims to those supported by the results. Authors should explicitly state that they have been unable to measure CRK5 activity and therefore cannot unequivocally state that it requires SOC2 or that the complex behaves as a cyclin-dependent kinase, despite the hints from sequence similarity; cyclin-dependent activity needs to be demonstrated for it to be considered a bonafide CDK. Moreover, authors should not assume that changes in phosphorylation can be directly attributed to CRK5 activity because too many changes are occurring during exflagellation (or are not occurring when CRK5 is absent) to unequivocally attribute them as direct phosphorylation events.

3) Further discussion of the AID system: The authors should discuss that the spindle formation is already deficient in the AID-tagged strain (at the level of the KO) and therefore not further diminished by auxin treatment. Please include the following critical experimental details: Was the timing of CRK5 degradation following auxin treatment determined? Were gametocytes pre-treated with auxin in the pa experiments and, if so, for how long? Presumably, it was not 1 min pa activation combined with 1 min auxin induction, which would simply be not enough to eliminate CRK5.

---

## [Author Response]

Essential revisions:The major concerns of the reviewers fall into three general categories:1) Complex formation and mass spectrometry.a) The mass-spectrometry experiments are poorly explained and the tables poorly annotated. There is no mention in the text of where TMT labelling was used, numbers of replicates, numbers of proteins identified etc. Fold-change significance should ideally be based on median standard deviation of the dataset, not an arbitrary cutoff. Little information is provided for the IP datasets, which are the foundation of the CSC complex and it seems critical to include control IPs against parasites of similar stages lacking an epitope tag, to establish the background of non-specific binding.

We thank the reviewers for pointing to a lack of clarity regarding the mass spectrometry experiments. We agree methods and results were not clearly explained in the initial version. Therefore, the presentation of the results did not reflect well the richness of the datasets.

We are now indicating in the Results section which time points have been analysed for SOC2, CRK5 and CDKrs IPs. We have removed the Venn diagram representing a qualitative analysis all IPs (previous Figure 3C). Instead, we now show a semi-quantitative PCA analysis of SOC2, CRK5, CDKrs IPs at 1 min post-activation. We are comparing these results with IPs of CDPK4-HA and its substrates SOC1-HA and SOC3-HA as well as of MCM5-HA. This analysis shows that the three proteins cluster together consistent with the formation of a complex.

We omitted to state that the IP data was already compared to a control IP against gametocytes lacking an epitope tag. All proteins identified in this control were already filtered out in the previous version. We have now included this information and show the control results in Supplementary file 4. Proteins identified by less than 2 peptides were also filtered out in this table. We have also included a WT-GFP control in Supplementary file 4.

Label-free comparison of peptide and phosphopeptides in SOC2 or CRK5 immunoprecipitates. Label-free comparison of peptide and phosphopeptides in SOC2 or CRK5 immunoprecipitates during the first minute of gametogony was determined using the “Minora Feature Detector” Node of pProteome dDiscoverer 2.2 (Thermo Fisher Scientific). Peptide-spectrum matches were validated using Percolator validator node with a target FDR of 0.01 and a Delta Cn of 0.5. For label-free quantification, features and chromatographic peaks were detected using the “Minora Feature Detector” Node with the default parameters. PSM and peptides were filtered with a false discovery rate (FDR) of 1%, and then grouped to proteins with again an FDR of 1% and using only peptides with high confidence level. Both unique and razor peptides were used for quantitation and protein abundances are calculated by summing samples abundances of the connected peptides group. The abundances were normalised on the “Total Peptide Amount” and then “The pairwise Ratio Based” option was selected for protein ratio calculation and associated p-values were calculated with an ANOVA test based on background. We now indicate this information in the main text and detail the analysis in the Materials and methods section.

We used TMT labelling for the proteomics and phosphoproteomics comparison of WT and CRK5-KO. Each of the five time points was analysed with two independent biological replicates, i.e. from purified gametocytes isolated from different mice. We now indicate this information in the Results and Materials and methods sections.

We did not use any cut-off to determine fold-change significance of proteomics and phosphoproteomics analyses. We only based our selection criteria on Benjamini-Hochberg adjusted p values (<0.05). We agree the grey areas in Figure 2B was misleading and suggested we used an arbitrary cut-off -0.5<Log_2_FC<0.5. The standard deviation ranged from 0.28 to 0.39 (proteome) and from 0.24 to 0.27 (phosphoproteome). As no significantly regulated sites with Benjamini-Hochberg adjusted p values <0.05 were below a Log_2_FC of 0.5 this does not affect our previous analysis.

b) IPs for CRK5, SOC2, and CDKrs all recovered the other putative complex components, but also several members of the MCM and ORC complexes. The authors should more precisely define the complex arrangement, since CDK-cyclin complexes canonically comprise two components despite their numerous associations throughout the cell. An experiment establishing the stoichiometry of components, or their hierarchy of association should aid in defining the putative complex. As described below, linking the complex composition to its function through a more detailed analysis of the CRK5 C-terminus could fulfil concerns about the complex, as well as about its activity.

As indicated in the previous point, we removed the Venn diagram representing a qualitative analysis of all IPs (previous Figure 3C). Instead, we now show a semi-quantitative PCA analysis of IPs of SOC2, CRK5, CDKrs at 1 min post-activation in modified Figure 3C. We are comparing these results with IPs of CDPK4-HA and its substrates SOC1-HA and SOC3-HA as well as of MCM5-HA that were all previously shown to be nuclear proteins. This analysis shows that SOC2, CRK5, CDKrs cluster together, consistent with the formation of a complex.

2) Complex activity.The authors should establish that the complex is either activated during gametogony or dependent on SOC2 for activity. In vitro kinase assays could be performed as described by Dorin-Semblat et al., 2013. Alternatively, the activity of complexes isolated from the WT and SOC2-defficient parasites, or before and after activation, could help establish the conditional activation of the complex.

The conditional degradation of CRK5 in terminally differentiated gametocytes minutes before activation together with minor changes at the phosphoproteome level observed in CRK5-KO cells prior to activation strongly suggest an activation of CRK5 following gametocytes stimulation. However, we agree we cannot rule out that CRK5 or the complex itself could possibly be activated prior to gametocyte stimulation or independently of SOC2. In vitro kinase assays to assess SOC2-dependent activation of CRK5 is an excellent suggestion. We have tried to reproduce the in vitro kinase assays presented by Dorin-Semblat et al., 2013, but our attempts indicate that such experiments may not be as trivial as suggested.

In a first time, we have tried to perform in vitro kinase assays using CRK5-GFP immunoprecipitates from activated gametocytes. Of note, our MS analysis of such immunoprecipitates indicate the presence of other kinases in these immunoprecipitates including CDPK4, MAPK2, NEK4 possibly masking CRK5 activity. Kinase assays were performed using Kinase-Glo Assay kit as per manufacturer instructions. As a control, we used immunoprecipitates from MyoB-GFP, a protein with no kinase or ATPase function expressed in gametocytes. Kinase reactions containing 40 mM Tris-HCl (pH 7.5), 20m M MgCl_2_ were performed at 10 μM ATP and 50 μM Histone H1 with different dilutions of immunoprecipitates in solid white 96-well plates in 50μl kinase reaction buffer. MyoB-GFP was used as negative control whereas recombinant Protein kinase A (PKA, 10 units) was used as positive control. Following the kinase reaction, an equal volume of Kinase Glo Max Reagent (50μl) was added. Luminescence was recorded on a GloMax 96 Microplate Luminometer after ten minutes. Unfortunately, the kinase activity assays did not show any specific kinase activity for CRK5-GFP (Author response image 1) when compared to MyoB-GFP.

**Author response image 1. sa2fig1:** CRK5 Kinase activity assay using Kinase-Glo Assay kit. Kinase reactions containing 40 mM Tris-HCl (pH 7.5), 20 mM MgCl_2_ were performed at 10 μM ATP, 50 μM Histone with different dilutions of kinases for 1 hour. X-axis represents: Control (-) – No kinase, Substrate (-) – No substrate, 15 and 30 – quantity of CRK5-GFP and MyoB-GFP beads (μl), PKA – 10 units of Protein kinase A. Error bars represent standard deviations from the mean; two independent biological replicates.

We then tried to express recombinant CRK5, SOC2 and CDKrs from *E. coli* as performed in Dorin-Semblat et al., 2013. As our results indicate that the C-terminus extension of CRK5 is essential for its function in vivo, we also decided to express full-length CRK5. The SOC2 (PfCyc2) expressed in Dorin-Semblat et al., 2013, only encompasses 391 amino acids of the cyclin box (see Merckx et al., 2003) which only represents a fraction of full length SOC2. Here we have cloned the full sequence encoding the 1314 amino acid protein that is detected to interact with CRK5. Similarly, we aimed at expressing and purifying full-length CDKrs. Codon-optimised sequences for *E. coli* were obtained from GeneArt with N-terminus 6xHis tags. After full-length sequencing of each construct, we attempted overnight expression in *E. coli* BL21 cells. Although a strong expression of each protein was achieved, none of them was soluble preventing further purification steps (Author response image 2). This suggests either that the proteins aggregate because they are not properly folded, or that they interact with membranes and sediment upon centrifugation after cell lysis.

**Author response image 2. sa2fig2:** Expression of CRK5, CDKrs and SOC2 in *E. coli* BL21 cells. Overnight IPTG-induced expression of CRK5, CDKrs or SOC2 in *E. coli* BL21 (DE3) cells at 16ᵒC. Cell pellets were resuspended in 60 ml of lysis buffer (50 mM Tris pH 7.4/ 200 mM NaCl/ 3 mM 2-ME – fresh) and lysed by two passages through a French Press at 1,000 psi, followed by sonication for 20 sec, 20 % duty cycle, power 5. Cell debris and membranes were pelleted by centrifugation (30'000 g, 35 min, 4 °C). Soluble (supernatant) and non-soluble (pellet) fractions were denatured and analysed by SDS-PAGE. Expected MWs are: CRK5: 85 kDa, CDKrs: 35 kDa, and SOC2 158 kDa. Please also note that lower MW are observed for CRK5 and SOC2.

We thus decided to express each of these proteins tagged with N-terminal 10xHis and 2xStreptavidin tags in Sf9 insect cells. Baculoviruses expressing the protein of interest were generated from a modified pFastBac vector following the "Bac to Bac" protocol. Briefly, a bacmid encoding the gene of interest was isolated and transfected into fresh Sf9 insect cells. After 7 days of infection, first generation (P1) viruses were harvested and directly used to infect fresh Sf9 cells to produce highly infective second-generation viruses (P2). 4500 ml of Sf9 insect cells were infected with 4 ml of P2 viruses and grown for 66 h at 27°C. Cells were harvested by centrifugation at 3000 g for 12 min. Initial tests indicated low expression of the three proteins.

We reasoned that co-expression of all three proteins could maximise our chances of expression and stabilisation. To this aim, we used a MultiBacmid expression system that enabled the expression of many proteins and protein complexes that have been intractable before. Using the same expression protocol, an improved expression was observed. To lyse the cells, the pellet was resuspended in 60 ml of lysis buffer (50 mM Tris pH 7.4/ 200 mM NaCl/ 3 mM 2-ME – fresh). Cells were lysed by two passages through a French Press at 1000 psi, followed by sonication for 20 sec, 20 % duty cycle, power 5. Cell debris and membranes were removed by centrifugation (30'000 g, 35 min, 4 °C). However, proteins were again poorly soluble (Author response image 3A).

In parallel, we added the signal sequence for honeybee melittin, a highly expressed and efficiently secreted protein, to direct expression of recombinant proteins through the secretory pathway to the extracellular medium. Mellitin improved the expression or stabilisation of each of the three proteins; however, none of them could be recovered from the culture supernatant (data not shown) or could be solubilised efficiently from cell lysates (Author response image 3A). Chromatography affinity purification on His-Tap resin of solubilised proteins allowed significant enrichment for CDKrs but not for CRK5 or SOC2 although the three proteins could be detected by western blot after purification (Author response image 3B and 3C). Similar results were obtained using STREP-Tactin resin (data not shown).

**Author response image 3. sa2fig3:** Expression and tentative purification of SOC2, CRK5 and CDKrs in Sf9 cells. (**A**) Expression of SOC2, CRK5 and CDKrs N-terminally fused to the mellitin signal peptide, as well as 10xHis and 2xStreptavidin tags individually or in co-expression (all). Cell pellets were resuspended in 60 ml of lysis buffer (50 mM Tris pH 7.4/ 200 mM NaCl/ 3 mM 2-ME – fresh) and lysed by two passages through the French Press at 1000 psi, followed by sonication for 20 sec, 20 % duty cycle, power 5. Cell debris and membranes were pelleted by centrifugation (30'000 g, 35 min, 4°C). Soluble (Sol) and non-soluble (WCL, whole cell lysate) fractions were denatured and analysed by SDS-PAGE. (**B**) The supernatant containing soluble proteins was applied to a 5 ml Strep-TACTIN column and washed with 40 ml lysis buffer (Flow). WCL, flow and proteins retained on beads were analysed by SDS-PAGE stained with coomassie. C. Western blot analysis of proteins bound to beads.

In conclusion, we agree that the data presented in Dorin-Semblat et al., 2013, is attractive. They are however not as trivial as suggested in this paper and we are currently unable to reproduce them with our experimental setting. We have thus toned-down our statements on the potential SOC2-dependent regulation of CRK5 in the Discussion.

3) Complex Function.There is insufficient evidence to establish the CRK5 control of both S and M phases (Abstract) and the authors should tone down their conclusions regarding the precise function of the complex. It is expected that blocking DNA replication will not wholly block preparation for mitosis. Concurrent events are likely controlled in parallel or partially independent ways. Furthermore, CRK4 is involved in S/M phase progression at the RBC stage and may also play a role in gametogenesis (Ganter et al., 2017); authors consider this possibility (Discussion paragraph two) but do not include it in their final statements (Abstract). Furthermore, changes in phosphorylation are not sufficient to establish regulation, especially since the possibility that the block in cell cycle progression has not caused a change in total protein abundance.

We fully concur with this comment. We have been over-enthusiastic in interpreting the data presented in this manuscript probably because we have obtained in parallel to this project more and more data suggesting an uncoupling between S and M-phases during gametogony. We have toned down our interpretation regarding a possible role of CRK5 during M-phase in the Results and Discussion. As this only represents an anecdotal part of the manuscript, we removed any mention of a possible role in M-phase regulation from the Abstract.

We agree that changes in phosphorylation are not sufficient or necessary to establish regulation. However, we observe CRK5-dependent phosphorylation of multiple components of the pre-replicative complex at CDK consensus motifs and show a strong defect in DNA replication, all in a very narrow time window, which strongly suggest an important role of phosphorylation in establishing regulation. Although we cannot rule out a kinase-independent role of CRK5, it is likely that CRK5 kinase activity is important to regulate DNA replication and more likely initiation of DNA replication. We only mention this possibility in the Discussion and emphasize that future studies will have to (i) directly link CRK5 activity and phosphorylation of the pre-replicative complex and (ii) to experimentally determine the functional relevance of this phosphorylation events.

[Editors' note: further revisions were suggested prior to acceptance, as described below.]

Essential revisions:1) Proteomics: Authors should use the same gene IDs in the source data files as in the text. All columns should be labelled and a legend should explain how the values for that column were derived. Log_2_FCs for each phosphorylation site should be clearly visible with most important datapoints (localisation score, quality of MS assignment, peptide identified) for each timepoint. The authors should comment and discuss why some of the proteins are already differentially phosphorylated at t=0 (for example ORC2).

We have reformatted the tables related to the phosphoproteome and immunoprecipitation analyses so that the gene IDs are consistent between the tables and the text.

We have changed the FC to Log_2_FC.

All the other requested information was already present in the table. However, we agree that there was an excess of less informative columns and it was difficult to extract relevant information. We have removed less relevant information, added the protein name and description and added the possibility to filter p-values to query the table more easily.

We believe that the difficulty to query this table may have confused the reviewers as there are only two sites significantly less phosphorylated at t=0 (adjusted p-value <0.05): CRK5-S582 and PBANKA_1431400-S1053. To avoid any confusion we have added tabs showing peptides significantly regulated for each time point.

2) CDK nomenclature and function: The authors are meticulous throughout the text stating that their conclusions are suggestive of CDK function, but the final statements of the Introduction or Discussion are not supported by the results e.g. "This study indicates that CRK5, despite its divergence from canonical CDKs, acts as a genuine CDK by binding a cyclin and phosphorylating classical CDK motifs on proteins that are targeted by conventional CDKs and involved in the initiation of DNA replication in human and yeast." The reviewers instruct the authors to limit their claims to those supported by the results. Authors should explicitly state that they have been unable to measure CRK5 activity and therefore cannot unequivocally state that it requires SOC2 or that the complex behaves as a cyclin-dependent kinase, despite the hints from sequence similarity; cyclin-dependent activity needs to be demonstrated for it to be considered a bonafide CDK. Moreover, authors should not assume that changes in phosphorylation can be directly attributed to CRK5 activity because too many changes are occurring during exflagellation (or are not occurring when CRK5 is absent) to unequivocally attribute them as direct phosphorylation events.

We thank the reviewers for pointing to these over interpretative statements. We have now changed the above-mentioned statement as follows:

End of Introduction: “Here, *we provide evidence suggesting* CRK5 is part of a unique and divergent CDK/cyclin complex required for progression through male gametogony and essential for parasite transmission.“ (emphasis added).

Results: “Altogether this phosphoproteomic *shows that components of the pre-replicative complex are phosphorylated on a canonical CDK motif in a CRK5-dependent manner suggesting that CRK5 functions as a CDK*.” (emphasis added)

Conclusion: *“Plasmodium* species lack clear CDK orthologues (Cao et al., 2014) that are usually highly conserved across Eukarya and indispensable in yeasts and humans (Malumbres, 2014). This study indicates that CRK5, despite its divergence from canonical CDKs, *possibly* acts as a CDK by binding a *divergent* cyclin and phosphorylating classical CDK motifs on proteins that are targeted by conventional CDKs and involved in the initiation of DNA replication in human and yeast. However, it remains difficult to identify a specific direct functional orthologue due to functional similarities with multiple CDKs and even non-CDK cell-cycle kinases, and it is tempting to speculate that CRK5 and SOC2 cover the functional space of multiple cell-cycle regulators in other organisms, to satisfy the specificities of the *Plasmodium* cell cycle. *Future functional analyses will have to confirm that CRK5 activity is regulated in a SOC2-dependent manner and to demonstrate the direct phosphorylation of the pre-replicative complex by CRK5*.” (emphasis added)

3) Further discussion of the AID system: The authors should discuss that the spindle formation is already deficient in the AID-tagged strain (at the level of the KO) and therefore not further diminished by auxin treatment. Please include the following critical experimental details: Was the timing of CRK5 degradation following auxin treatment determined? Were gametocytes pre-treated with auxin in the pa experiments and, if so, for how long? Presumably, it was not 1 min pa activation combined with 1 min auxin induction, which would simply be not enough to eliminate CRK5.

The information was stated in the Results section but we have now added the procedure in the Results section and provided more details in the Materials and methods section:

Results: “Addition of the AID/HA tag to the CRK5 C-terminus imposed a significant fitness cost, with a 2-fold decrease in exflagellation in the absence of auxin, but importantly, depletion of CRK5-AID/HA by *one hour of* auxin treatment of mature gametocytes *prior to activation* resulted in a dramatic reduction in exflagellation (Figure 1H).” (emphasis added)

Materials and methods: “To induce CRK5-AID/HA degradation, 1 mM auxin dissolved in ethanol *(0.2% final concentration)* was added to purified gametocytes for one hour *prior to activation by XA.”* (emphasis added)